# Sc-compReg enables the comparison of gene regulatory networks between conditions using single-cell data

Zhana Duren[1,2,3,7], Wenhui Sophia Lu[3,7], Joseph G. Arthur[4], Preyas Shah[4], Jingxue Xin [3,5], Francesca Meschi[4], Miranda Lin Li[5], Corey M. Nemec [4], Yifeng Yin[4] & Wing Hung Wong [3,5,6✉]

The comparison of gene regulatory networks between diseased versus healthy individuals or between two different treatments is an important scientific problem. Here, we propose sc-compReg as a method for the comparative analysis of gene expression regulatory networks between two conditions using single cell gene expression (scRNA-seq) and single cell chromatin accessibility data (scATAC-seq). Our software, sc-compReg, can be used as a stand-alone package that provides joint clustering and embedding of the cells from both scRNA-seq and scATAC-seq, and the construction of differential regulatory networks across two conditions. We apply the method to compare the gene regulatory networks of an individual with chronic lymphocytic leukemia (CLL) versus a healthy control. The analysis reveals a tumor-specific B cell subpopulation in the CLL patient and identifies TOX2 as a potential regulator of this subpopulation.

[1] Center for Human Genetics, Clemson University, Greenwood, SC, USA. [2] Department of Genetics and Biochemistry, Clemson University, Clemson, SC, USA. [3] Department of Statistics, Stanford University, Stanford, CA, USA. [4] 10X Genomics, Inc., Pleasanton, CA, USA. [5] Bio-X Program, Stanford University, Stanford, CA, USA. [6] Department of Biomedical Data Science, Stanford University, Stanford, CA, USA. [7] These authors contributed equally: Zhana Duren, Wenhui Sophia Lu. ✉email: whwong@stanford.edu

The recent development of genomics technology enables us to measure genomics features at the single-cell (sc) level; for example, scRNA sequencing (scRNA-seq)[1] enables transcription profiling, scATAC sequencing (scATAC-seq)[2] identifies accessible chromatin regions, and sc-bisulfite sequencing[3] measures DNA methylation, all at the single-cell level. The advent of single-cell (sc) genomics has raised many interesting questions in data analysis. Figure 1A summarizes some common experimental designs and analysis tasks. For example, when a scRNA-seq sample[1] is obtained from a heterogeneous cell population, we may want to identify clusters of cells with distinct expression profiles[4,5] (Fig. 1A(i), upper left). With two scRNA-seq samples from two different cell populations (say, corresponding to two treatment conditions), then in addition to finding cell clusters, we may ask whether a cluster in one of the samples can be linked to one or more clusters in the other sample[4,5] (Fig. 1A(ii), lower left). Many methods have been proposed for the analysis of such scRNA-seq data sets[6–8].

In addition to gene expression, one may be interested in analyzing gene regulation, that is, to identify the transcriptional factors (TFs) that may regulate the expression of a target gene (TG), and the cis-REs relevant for the regulation. To study gene regulation in a cell population, a widely used approach is based on paired single-cell expression and accessibility analysis, where, scRNA-seq and scATAC-seq[2] experiments were performed on two different samples of cells from the same cell population (Fig. 1A(iii), upper right). Duren et al.[9] have discussed how to infer subpopulation-specific gene expression and chromatin accessibility profiles. For each subpopulation, the availability of TG expression and RE accessibility, together with TF-motif matching scores on accessible REs, allowed detailed inference of subpopulation-specific regulatory relations. Recently, several other methods like SOMatic[10], scAI[11], and MAESTRO[12] are developed to integrate scRNA-seq with scATAC-seq. Finally, this paired approach can also be used for the comparative regulatory analysis of two cell populations (Fig. 1A(iv), lower right), raising many novel challenges in the data analysis (see below). Our aim in this work is to address these challenges.

The main methodological question is how to identify differential regulatory relations based on four single-cell data sets (1 scRNA-seq + 1 scATAC-seq, from each of two populations). Furthermore, this comparison should be done in a subpopulation-specific manner. For example, in the Chronic Lymphocytic Leukemia (CLL) example to be discussed later, the population of primary bone marrow mononuclear cells (BMMC) is composed of many distinct subpopulations such as B cells, T cells, monocytes, NK cells, etc. Comparisons involving different cell types, say B cells from CLL patient versus NK cells from a healthy donor, are not of interest because there are already too many regulatory differences between the two cell types. Instead, we are mainly interested in detecting differences of regulatory relations between subpopulations of the same type (linked subpopulations), such as CLL B cells versus healthy B cells, CLL T cells versus healthy T cells, etc. Several methods have been developed to detect the differential network based on bulk RNA-seq data[13–15]. However, detecting the differential regulatory network by integration and comparison of scRNA-seq and scATAC-seq is a novel problem and it is challenging because whether a TF is involved in the differential regulation of a TG may depend in complex ways on the expression of the TF and the accessibility of REs that may mediate the activity of the TF on the TG (see Fig. 2).

In this paper, we present a new statistic for testing differential regulatory relations across two linked subpopulations. Although our statistic is derived as a likelihood ratio statistics, the standard Chi-square distribution is not appropriate as its null distribution.

A

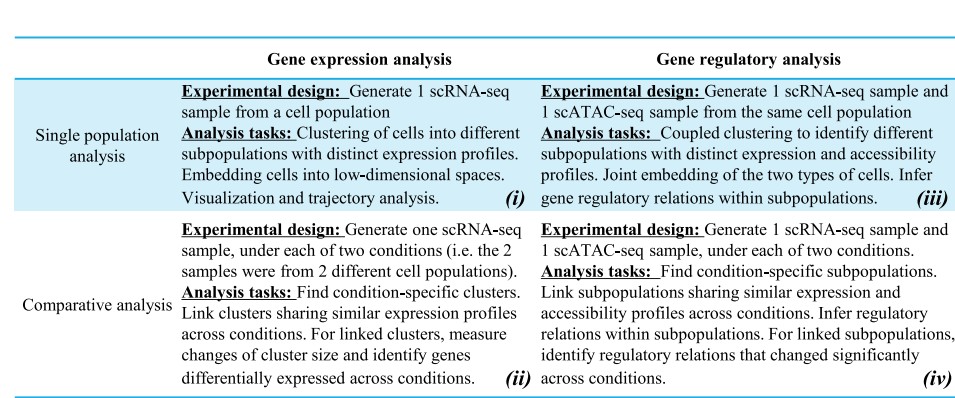

B

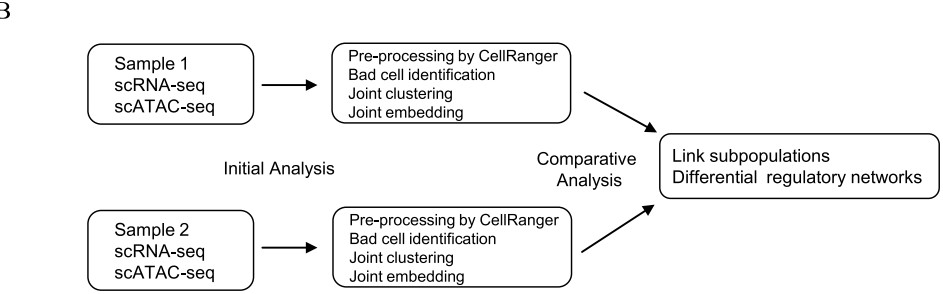

**Fig. 1 Schematic overview of comparative regulatory analysis. A** Summary of four types of single-cell genomic analysis. Left column: expression analysis by sc-RNA-seq. Right column: regulatory analysis based on paired scRNA-seq and scATAC-seq. Top row: analysis under one condition. Bottom row: comparative analysis. **B** Comparative regulatory analysis pipeline for scATAC-seq and scRNA-seq data from two conditions.

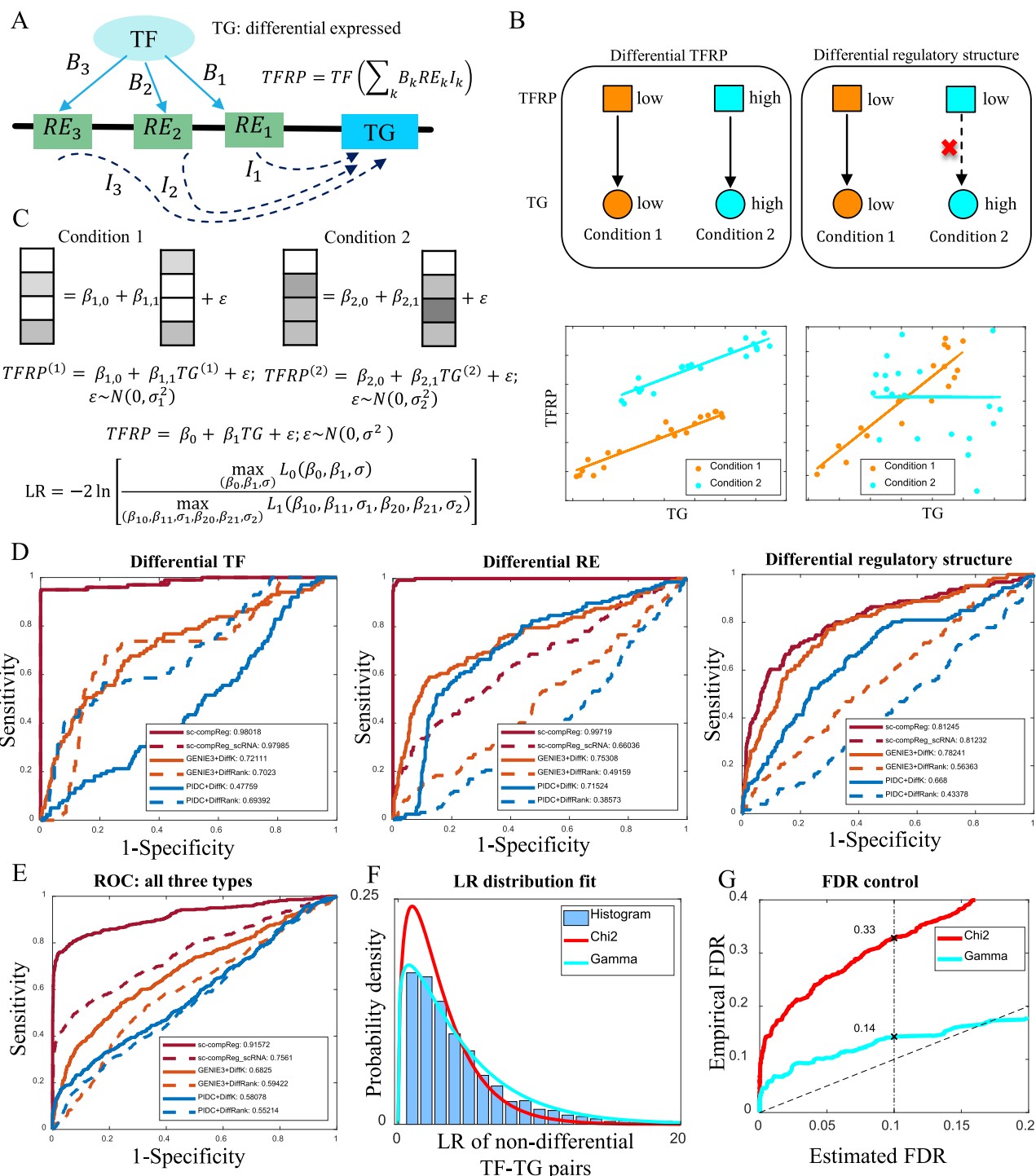

**Fig. 2 Concept and validation of sc-compReg. A** Concept of TF regulatory potential (TFRP). Here TF indicate TF expression, $RE_k$ is the accessibility of the $k$-th RE, $B_k$ indicates the TF-motif matching strength in the $k$-th RE, and $I_k$ is an interaction strength for whether the $k$-th RE is linked to the TG in the subpopulation-specific regulatory network (constructed in a preceding step in the sc-compReg pipeline. **B** Two types of differential regulations: differential TFRP (left) and differential regulatory structure (right). **C** Comparison of the conditional distribution of TFRP given TG expression by the likelihood ratio, for detail please see "Statistical test for differential regulatory relations" in the "Methods" section. **D** ROC curves of differential regulatory detection on three simple scenarios of simulation data: differential TF expression, differential RE accessibility, and differential regulatory network structure. For simulation detail please see "Differential regulatory network data simulation" in the "Methods" section. **E** ROC curves of differential regulatory detection on simulation data containing all three types of differential regulation. **F** Comparison of likelihood ratio distribution with chi-square distribution and Gamma distribution. **G** Comparison of estimated FDR with the empirical FDR.

We will introduce methods for *p*-value computation and FDR control, and present simulation results to validate the methodology. This is the key methodological contribution of this paper. Our second contribution is the development of the software *sc-compReg* for single-cell comparative gene REGulatory analysis. Before one can test for differential regulatory relations across linked subpopulations, many intermediate analyses must be performed on the single-cell data sets to identify the linked subpopulations (Fig. 1B). Thus, in its "Initial Analysis" step, *sc-compREG* has incorporated many methods originally developed for comparative expression analysis or single population regulatory analysis, as reviewed in Fig. 1A. This allows our software to be used as a stand-alone package that takes scRNA-seq and scATAC-seq count data as input and provides a comprehensive set of results, including consistent cluster labels for cells from both scRNA-seq and scATAC-seq, the joint embedding of the two types of data for visualization, subpopulation-specific expression and accessibility profiles, inference of differential TF-TG relations, and differential regulatory networks. These functionalities are provided in *sc-compReg* without the need for the user to engage external software for intermediate steps in the analysis pipeline. The sc-compReg software is freely available (https://github.com/SUwonglab/sc-compReg). Before we perform comparative regulatory analysis, we need an important initial analysis step to obtain the clustering label for each of the single cells and match the subpopulations across two conditions. To ensure the robust performance of the initial analysis, we developed a rigorous validation methodology, which may be of independent interest in the evaluation of single-cell analysis methods. To validate a single-cell analysis method, we apply it to scRNA-seq and/or scATAC-seq samples from a heterogeneous population with constituent subpopulations that are already analyzed by bulk sample RNA-seq and/or ATAC-seq. We use the bulk sample profiles to compute surrogate "ground truth" labels to the cells in our samples. These labels can then be used to validate the clustering, embedding, and subpopulation matching in the initial analysis step of sc-compReg. Applying this strategy on single-cell data from a BMMC population, we showed that the initial analysis step of the sc-compReg pipeline is well-validated. Finally, as an illustration of our method on real data, we apply it to the comparative analysis of scRNA-seq and scATAC-seq data from a CLL donor versus that from a healthy control. Our analysis reveals a B cell subpopulation specific to the CCL donor, and identifies TOX2 as a key regulator within that population.

## Results

**A statistical method for detecting differential regulatory relations**. We propose a statistical method for comparative analysis of the gene regulatory networks in two linked subpopulations. For each subpopulation, its gene regulatory network is just the collection of gene pairs (TF, TG) where TF is a transcription factor that regulates the target gene TG. We call such a (TF, TG) pair a regulatory relation. To identify differential regulatory relations, first, we use a *t*-test to identify TGs that are differentially expressed between the two conditions. Once the differential TGs are known, the key challenge is to identify the TFs that are contributing to the differential expression of these TGs. If only scRNA-seq data is available, then it is natural to identify such TFs by searching for TFs whose expression correlation with the differential TG is significantly different in the two linked subpopulations. We refer to this as the baseline method for testing for TFs that regulate differential TGs.

By incorporating accessibility data on regulatory elements (RE), we design a new method more sensitive than the baseline method. To do this, we first construct a numerical index to represent the regulatory potential of a TF on a TG. This index, which we call transcription factor regulatory potential (TFRP), is a cell-specific index defined as the product of the (cell-specific) expression of the TF and its regulatory potential on the TG, where the regulatory potential is calculated by integrating accessibility information from multiple REs that may mediate the activity of the TF to regulate the TG (Fig. 2A). We note that the accessibility information needed in the TFRP computation can be obtained from the subpopulation-specific RE accessibility profiles, which have already been estimated in the coupled-clustering step of sc-compREG (Fig. 1b).

Generally, the differential regulation of a TG by a TF may be due to one or both of the following mechanisms. (1) Changes in TFRP: the TF regulates the TG in both two conditions, but TFRP differs greatly between the two conditions. Either change in TF expression or RE accessibility would induce a differential TFRP (Fig. 2B, left). (2) Changes in regulatory network structure: TFRP in two conditions are similar, but the TF regulates the TG under one condition but the regulation is absent under the other condition (Fig. 2B, right). From Fig. 2B, it is seen that the conditional distribution of TFRP given TG will be changed when there is differential regulation. This suggests that we could detect differential regulations by testing for changes in this conditional distribution. To do this, we use simple linear regression to model the dependency of TFRP on TG. A likelihood ratio statistic (LR) is computed to test the null hypothesis that the model is the same across the two conditions (Fig. 2C). Although in theory, the LR statistics should follow approximately a Chi-square null distribution, we found that this is not a good approximation in our case. Instead of the Chi-square, we propose a null distribution based on fitting a Gamma distribution, to the lower quantiles of the likelihood ratios ("Methods").

To validate our method, we simulate scRNA-seq and scATAC-seq data under two conditions ("Methods"). In our simulation, we consider four different scenarios: the differential regulations are caused by (1) differentially expressed TFs only, (2) differentially accessible REs only, (3) differential TF−TG regulatory structure only, and (4) all of the above. We compare our method with a baseline method that only uses scRNA-seq information (i.e. by replacing the TFRP with TF expression), which is noted as sc-compReg_scRNA. Figure 2D shows the ROC curve of each method (as the LR statistic varies) under the first three scenarios. Our method achieves AUC of 0.9802, 0.9972, and 0.8124 respectively under scenarios 1, 2, and 3. As expected, our method and the baseline method perform similarly in scenarios 1 and 3 where the changes involve only TF expressions or TF−TG relations. In scenario 2, which involves changes in REs, our method offers a dramatic improvement over the baseline method. Finally, in scenario 4 (Fig. 2E) which includes all the three types of differential regulation, our method also performs much better than the baseline method (AUC of 0.9157 versus 0.7561). An alternative way of performing this analysis is inferring the gene regulatory networks in each condition first and then detect differential regulation by comparing the two networks. We also compare our method with any combinations of two gene regulatory network inference methods and two differential regulatory detection methods. Based on benchmarking literature of gene regulatory network inference from single-cell data[16] and differential regulatory network detection[14], we choose GENIE3[17] and PIDC[18] as methods for network inference, and choose diffK[19] and diffRank[20] as methods for network comparison. The results showed that our method outperforms all those methods (Fig. 2D, E).

To compute the *p*-value or control the false discovery rate, we need the distribution of the likelihood ratio under the null hypothesis (non-differential TF−TG pairs). Figure 2F shows the

null distribution of the likelihood ratio statistics. We found that the standard Chi-square distribution does not fit well, but a Gamma fitted to the lower quantiles can offer a good approximation (see "Methods"). If we choose the critical value by setting the estimated FDR (adjusted by Benjamini−Hochberg procedure) to be 0.1, the empirical false discovery rates achieved by the two methods are 0.33 for the Chi-square method and 0.14 for the Gamma method. Thus our method is reasonably calibrated while the Chi-square method severely underestimates the FDR (Fig. 2G). Overall, the simulation results show that our method detects differential regulation with high sensitivity and also offers good FDR control.

**Surrogate ground truths provide validation for methods in the initial analysis**. Before we can conduct comparative regulatory analysis, we must first conduct many initial analysis steps to obtain cluster labels of cells in the scRNA-seq and scATAC-seq samples (Fig. 1B). To ensure that we have implemented high-performance methods for these initial analysis tasks, we constructed surrogate ground truth labels for the cells by using cell-type-specific signatures from bulk data from pure cell types. The performance of our methods and competing methods are then assessed based on these surrogate ground truths. Here, we illustrate this strategy on scATAC-seq and scRNA-seq data from healthy donor BMMC.

First, we construct surrogate ground truth labels (see Supplementary Fig. S1A for names of the 13 cell types) for single cells based on cell type-specific signatures from FACS sorted bulk data (Fig. 3A, B, see "Methods"). The fraction of cells allocated to different cell types is highly consistent between scRNA-seq and scATAC-seq (Supplementary Fig. S1A, B). The surrogate ground truth is also largely consistent with t-SNE plots except for one group of scATAC-seq cells (Fig. 3A, located in the right upper corner of the scATAC-seq t-SNE plot) that contain all 13 cell types. This group of cells also exhibit other unusual features such as much higher read depth (8.63 fold), a low percentage of reads in peaks, and abnormal distribution of reads over chromosomes (details see Supplementary note S1, Supplementary Figs. S1, 2). These unusual features were exploited by our method to detect and remove the bad cells ("Methods"). Based on the surrogate ground truth labels for the scATAC-seq data, we also found that peaks called by Cell Ranger are superior to peaks called by the popular peak caller MACS2 (details see Supplementary note S2). Therefore we use peaks from Cell Ranger in the following analysis.

Next, we evaluate three approaches for obtaining consistent pairing of clusters for scRNA-seq and scATAC-seq, (1) clustering the two types of data separately (i.e. Cell Ranger) and then match the clusters manually, (2) clustering scRNA-seq first and then transfer the labels to scATAC-seq (i.e. Seurat V3), and (3) joint clusterings. To perform a joint clustering analysis of the scRNA-seq and scATAC-seq data, we added a parameter selection criterion to our previous CoupledNMF method (originally designed for Fluidigm data)[9], to optimize its performance for 10× Genomics Chromium Single Cell data ("Methods"). To evaluate clustering performance, we build a one-to-one mapping between the clusters and the true labels by bipartite graph maximum weighted matching ("Methods"). Figure 3C shows the accuracy of these three different clustering methods in scRNA-seq and scATAC-seq. In scRNA-seq clustering, Seurat and CoupledNMF have good performance but Cell Ranger fails to detect the CD8 minor population. In scATAC-seq clustering, CoupledNMF and Cell Ranger have good performance but Seurat suffers a twofold lower accuracy in Ery and NK compared to the other two methods. Overall, the CoupledNMF has the best

performance in the initial analysis step (details see Supplementary note S3). Thus we choose it as a default clustering method in the initial analysis of sc-compReg.

Based on the reduced dimension matrix from CoupledNMF, we implemented a joint embedding of scRNA-seq and scATAC-seq on the same t-SNE plot ("Methods"). In a joint embedding, it is desirable to see cells of the same ground truth label placed near each other regardless of their data type (scRNA-seq or scATAC-seq). This is indeed the case in our joint embedding results (Fig. 3D). Based on the t-SNE coordinates and the surrogate ground truths, we calculate the silhouette index for each cell to evaluate how similar the cell is to its own cluster compared to other clusters. The joint tSNE provides a significantly higher silhouette index than separate t-SNEs from Cell Ranger (sign rank test, $p = 5.7973e{-}171$ for scATAC-seq and $p = 5.7588e{-}08$ for scRNA-seq).

**Comparative analysis reveals tumor-specific B cell subpopulations in a chronic lymphocytic leukemia sample**. Having validated the methodology, we next test it in a real application, namely the comparison of single cells from the BMMC sample of a CLL donor to that from a healthy donor. The analysis of the healthy sample has already been discussed above. We apply the same analyses to the CLL sample and obtain seven clusters (Fig. 4A). The clustering results are highly consistent with the surrogate ground truth labels (Supplementary Fig. S8B). Based on surrogate ground truth labels, subpopulations 1, 2, and 7 are mapped to the B cells. We refer to them as subpopulations B1, B2, and B7 respectively. Subpopulations 3 and 6 are mapped to CD4 and Ery respectively. Subpopulation 4 is a mixture of NK and CD8. Subpopulation 5 is a mixture of Mono and some progenitor cells.

To systematically compare the two populations, first, we need to link the subpopulations across donors. Using surrogate ground truth labels, we can infer the dominant cell type in each subpopulation. Then, clusters can be linked across the CLL sample and the healthy sample according to cell type. In particular, the B1, B2, and B7 in the CLL sample are all linked to the single B cell subpopulation in the healthy donor. To enable the usage in applications where surrogate ground truth is not available, our method can also link subpopulations across donors based on the expression and accessibility profiles under the two conditions ("Methods"). In fact, for the CLL+healthy data, Fig. 4B shows that this profile-based linking is completely consistent with the surrogate ground truth-based linking. This suggests that our profile-based linking can be used as a general method to link subpopulations across conditions. From the expression and accessibility profile comparison, we also find that the variation between the healthy donor and CLL donor is small compared to the variation between cell types (Supplementary Fig. S8C). This underscores the need to link subpopulations across conditions before performing comparative regulatory analysis.

From the surrogate ground truth labels, we see a three-fold increase of the B cell population in the CLL donor compared to the healthy donor (Fig. 4C). We cluster B cell scRNA-seq data and scATAC-seq data by applying NMF with $K = 2{-}10$ and check the clustering stability (Supplementary Fig. S8A). The results show that there are three stable clusters in B cells from CLL donors. On both RNA-seq and ATAC-seq, the most distinct subpopulation among the three subpopulations in CLL (compared to healthy donor B cell) is B7, which suggests that B7 is a highly tumor-specific B cell subtype (Supplementary Fig. S9). We further compare those B cell subpopulations based on immunophenotype. Based on previous studies[21], a well-known surface

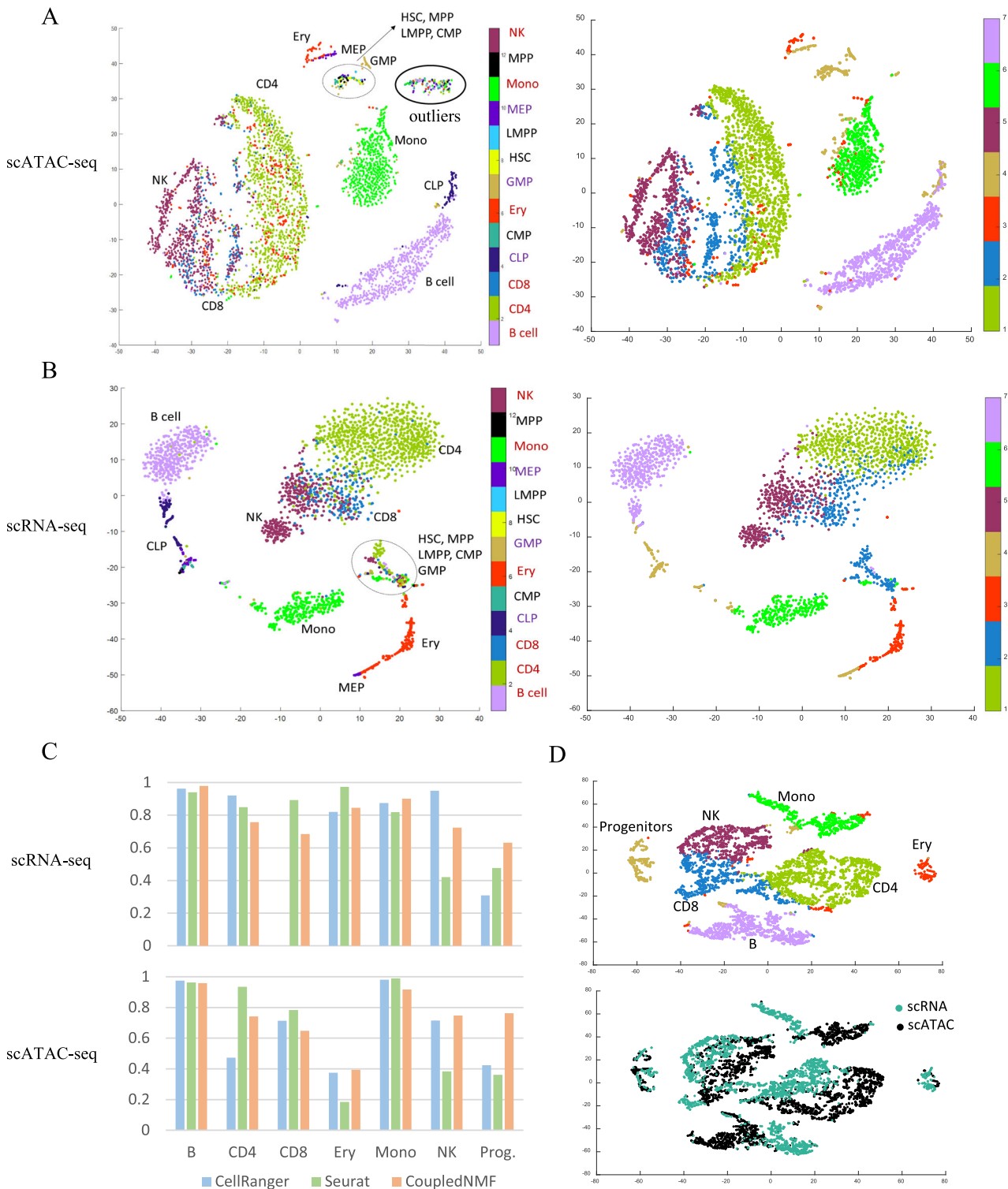

**Fig. 3 Validation of joint clustering and joint embedding on the normal sample. A** t-SNE plots of scATAC-seq data colored by true label (left) and clustering label from coupled NMF(right). **B** t-SNE plots of scRNA-seq data colored by true label (left) and clustering label from coupled NMF(right). **C** Comparison of the true label and clustering label for scATAC-seq (up) and scRNA-seq (down). **D** Joint embedding of scATAC-seq and scRNA-seq by using t-SNE on low dimension loading matrixes from coupled NMF.

marker of B cell CLL includes CD5, CD23, CD19, CD25, CD69, and CD71. In our data, *CD23*, *CD19*, and *CD69* are expressed in greater than 5% of the cells. Low expression of the other three markers maybe because of the drop-out of single-cell measurement. We systematically compare the pattern of markers *CD23*, *CD19*, and *CD69* in the four different B cell subpopulations.

Specifically, for each of the subpopulations (B1, B2, B7, and normal B cells), we first binarize the data as expressed or not expressed, and then calculate the distribution of cells among the eight types of expression patterns (i.e. *CD23+CD19+CD69+*, *CD23+CD19+CD69−*, *CD23+CD19-CD69−*, etc.). The degree of immunophenotype dissimilarity between two subpopulations

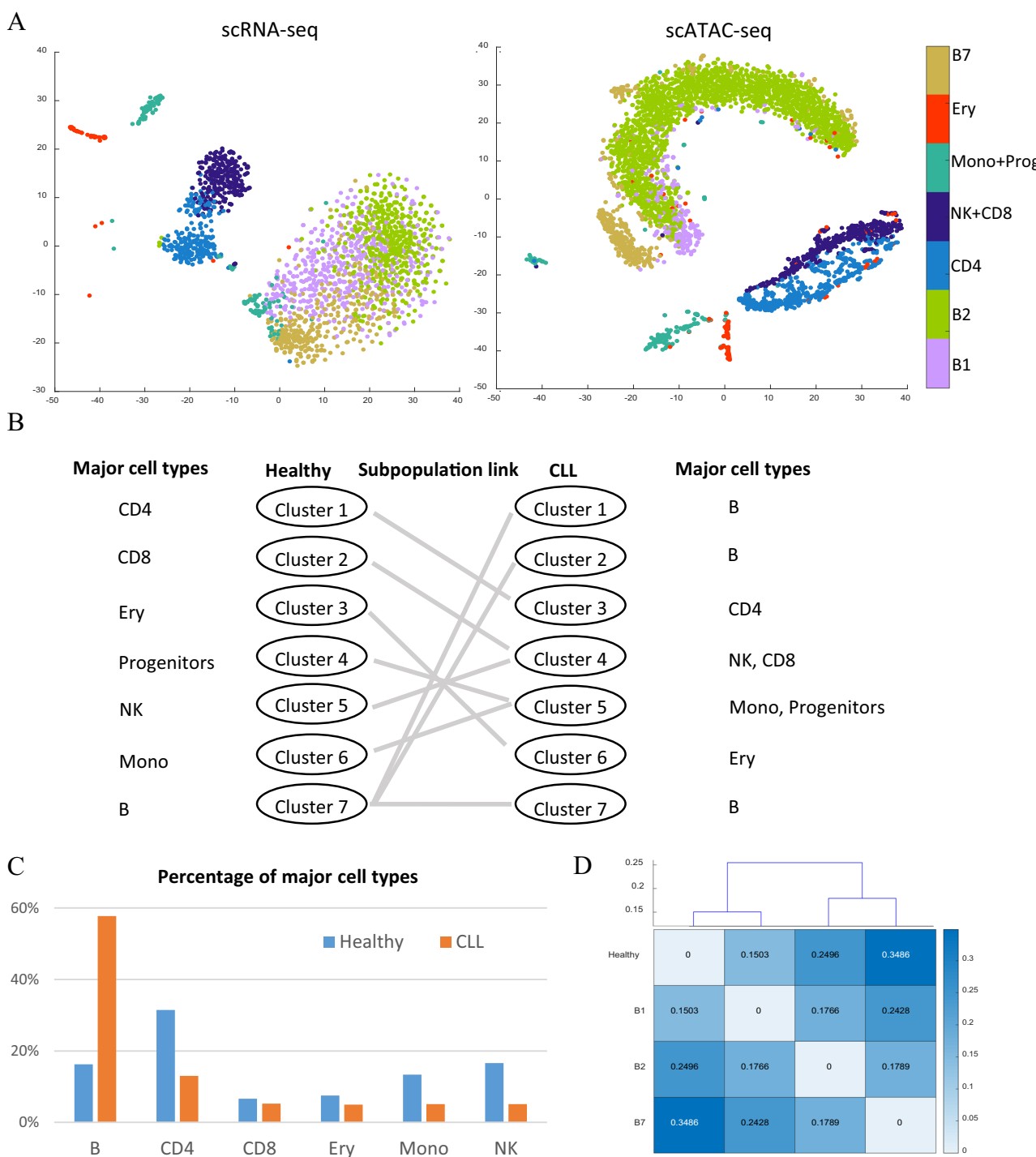

**Fig. 4 Comparative analysis of CLL with a healthy donor. A** tSNE plots colored with clustering label for scRNA-seq (left) and scATAC-seq (right) on CLL donor. **B** Cluster mapping between the healthy donor and the CLL donor. We find nine linked subpopulations based on our profile-based pairing method. The label in the bracket represents dominant surrogate ground truths. The result from profile-based pairing is completely consistent with the surrogate ground truths-based mapping. Please see "Linking of subpopulations across conditions" in the "Methods" section. **C** The fraction of cell types in healthy donor and CLL donor based on the constructed true labels. **D** Clustering of B cell subpopulation based on the joint pattern of markers CD19, CD23, and CD69. The numbers represent the Hellinger distance.

can be quantified by the Hellinger distance between the two corresponding distributions (Fig. 4D). It is seen that, among the three B cell subpopulations in CLL, B1 shows the highest similarity to normal B cells and B7 shows the least similarity to normal B cells. This is consistent with the global pattern from scRNA-seq and scATAC-seq (Supplementary Fig. S9). To

confirm this, we also did the copy number analysis based on copyKAT[22] ("Methods"). Supplementary Fig. S10A shows the B2 and B7 are 2.04 and 1.86 fold enriched (odds ratio) in the predicted aneuploid cells, while the odd ratio of healthy B cells and B1 are 0.22 and 1.22. We obtained 53, 75, and 66 loci which have significant copy number differences in B1, B2, and B7

compared to healthy B cells (Supplementary Data 1). All the significant loci in B1 are either overlapped with B2 or B7. B2 has 19 unique significant loci and B7 has 15 significant loci (Supplementary Fig. S10B). We examine whether those regions contain the previously identified copy number abnormalities regions for CLL like del13q, del11q, trisomy12, and del17p. We find all three B cell subpopulation contain del13q but only B7 contains del11q. The results from copy number analysis support that B1 is more similar to normal B cell and B7 is different from normal B cells.

We also compare the expression of the three B cell subpopulations in CLL donor with the healthy donor and identified 852, 1,247, and 1,206 CLL-specific genes for B1, B2, and B7 respectively. Taking the experimental validated TF−TG pairs[23,24] as ground truth, we compare our methods with other alternative methods and the results show our method has better performance than others (Supplementary Fig. S11). Among these CLL-specific genes, 351 are shared in all three subpopulations. The top-ranking subpopulation-specific genes are shown in Supplementary Fig. S12A (the common genes are removed). GO enrichment analysis (Supplementary Fig. S12B) identified several pathways/processes as enriched (Fold > 2, FDR < 0.05). Lipopolysaccharide-mediated signaling pathways, regulation of vascular smooth muscle cell proliferation, and regulation of toll-like receptor signaling pathways are enriched in the 351 common CLL-specific genes. B2 is enriched in myeloid cell development, indicating that B2 is in an earlier stage in development. B7 is enriched in cell cycle genes, which indicates that B7 may be a proliferative tumor subpopulation.

**Sc-compReg identifies TOX2 as a key regulator in a tumor-specific B cell subpopulation.** To explore the regulatory network difference in cancer versus healthy, we apply our sc-compReg method to compare each of B1, B2, and B7 with the healthy donor B cell and find three differential networks containing 686, 744, and 740 unique nodes and 237 common nodes shared between all three networks. From the common network, SOX4, MYB, KLF4, JUNB, and ID2 are identified as key TFs that may induce most of the differential expression between the healthy donor and CLL donor. From the three sets of unique nodes, we can find TFs that are specifically relevant to some subpopulations in CLL, e.g. ZNF415 is important for B1, FOXO4 is important for B2, TOX2 is important for B7 (Fig. 5A). Figure 5B shows the B7 specific network in which TOX2 (detail of TOX2 motif see Supplementary note S4) is a hub node and playing an important role.

The tumor-specific B cell subpopulation B7 has 855 specific genes (the common genes are removed) and the gene with the highest fold-change is *TOX2* (Supplementary Fig. S12A). Figure 5C shows the expression pattern of *TOX2* in healthy and CLL donors. *TOX2* is specifically expressed in the tumor-specific B cell subpopulation (B7) in CLL donor (44/369), but it is almost not expressed in the other two B cell subpopulations (5/522 and 0/535) in CLL donor, nor is it expressed in B cell subpopulation in the healthy donor (3/469). We examined the *cis*-elements near the *TOX2* gene in scATAC-seq data and identified four *cis*-elements that are specifically open in tumor-specific B cell subpopulation (Fig. 5D).

Next, we check the accessibility of the REs containing TOX motif (Supplementary Fig. S13), in B cell subpopulations in both healthy and CLL donors and find the tumor-specific B cell subpopulation (B7) has much higher accessibility than the other B cell subpopulations, which supports the idea that specific expression of *TOX2* induces a wide regulatory change (Fig. 5E). To examine whether *TOX2* is associated with CLL or not, we

check the expression of *TOX2* in B cells in two independent case-control studies[25,26]. Both studies show that *TOX2* is significantly highly expressed in the CLL group compared to the healthy control group (one-tailed two-sample *t*-test, *p*-value 0.0013, Fig. 5F, and Supplementary Fig. S14).

Based on gene expression and chromatin accessibility cluster profiles, we build a context-specific gene regulatory network for B7 by our previously developed PECA2 method[27]. Network analysis shows that target genes of TOX2 are remarkably highly enriched (4.5-fold, *p*-value 2.83E−09) in tumor-specific genes (Fig. 5G). GO enrichment analysis shows that TOX2's target gene is enriched in B cell activation, immune response, and leukocyte activation (Fig. 5H). Interestingly, TOX has recently been identified as a key regulator in exhausted T cells and cancer[28,29]. In our study, we did not find any difference in TOX gene expression between the CLL donor and the healthy donor in any cell type (Supplementary Fig. S15). Instead, our analysis identified a paralog of the *TOX* gene, *TOX2*, as an important regulator in one of this CLL patient's B cell subpopulations.

## Discussion

In this paper, we proposed a new test statistic for the comparison of gene regulation based on single-cell genomics data, developed the software sc-compReg, and suggested a new bulk data-based validation strategy for single-cell analysis. We applied our method to the comparison of CLL versus healthy control and identified TOX2 as a key regulator of transcriptional changes in a sub-population of B Cells identified in this CCL patient.

Comparison of gene regulatory networks from patients and healthy control would help us identify disease genes, dysfunctional regulations, and the underlying molecular mechanism of diseases. In the past, this type of analysis required bulk sample genomics data from a large sample. The comparison of patients and healthy control at the regulatory level has always been challenging because it is difficult to collect relevant tissues from a large number of clinical patient samples. The development of single-cell technologies has made it possible to learn regulatory relations from a single sample. In this work, we have shown that based on scRNA-seq and scATAC-seq, our method provides informative comparative regulatory analysis for one patient sample versus one healthy control. So we believe sc-compReg would be a useful tool for clinical sample gene regulatory analysis. Of course, the differential regulations discovered this way will need to be replicated by additional patients and controls. Thus an important next step will be to extend the approach to accommodate multiple donors under each of the two conditions to be compared. We will work on this extension in the near future when this type of data becomes available. We believe that using such single-cell analysis, the discovery of disease-associated changes in regulatory relations should be feasible using much fewer donors than is possible based on bulk sample data.

Our approach to comparative regulatory analysis can also be extended to handle more complex experimental designs. For example, recently it has been shown that the addition of bulk HiChIP data to sc-RNA-seq and scATAC-seq data can significantly enhance subpopulation-specific regulatory analysis[30]. In this situation, our TFRP-based test statistics can still be used to infer differential regulatory relations, as long as the computation of linked subpopulations in the initial analysis has considered the HiChIP data. We will extend our sc-compReg software to enable this type of analysis.

## Methods

**Single-cell ATAC.** Cryopreserved BMMCs from two donors were obtained from AllCells. One donor was a 93-year-old non-Hispanic white male with newly diagnosed chronic lymphocytic leukemia, and the other individual was healthy but

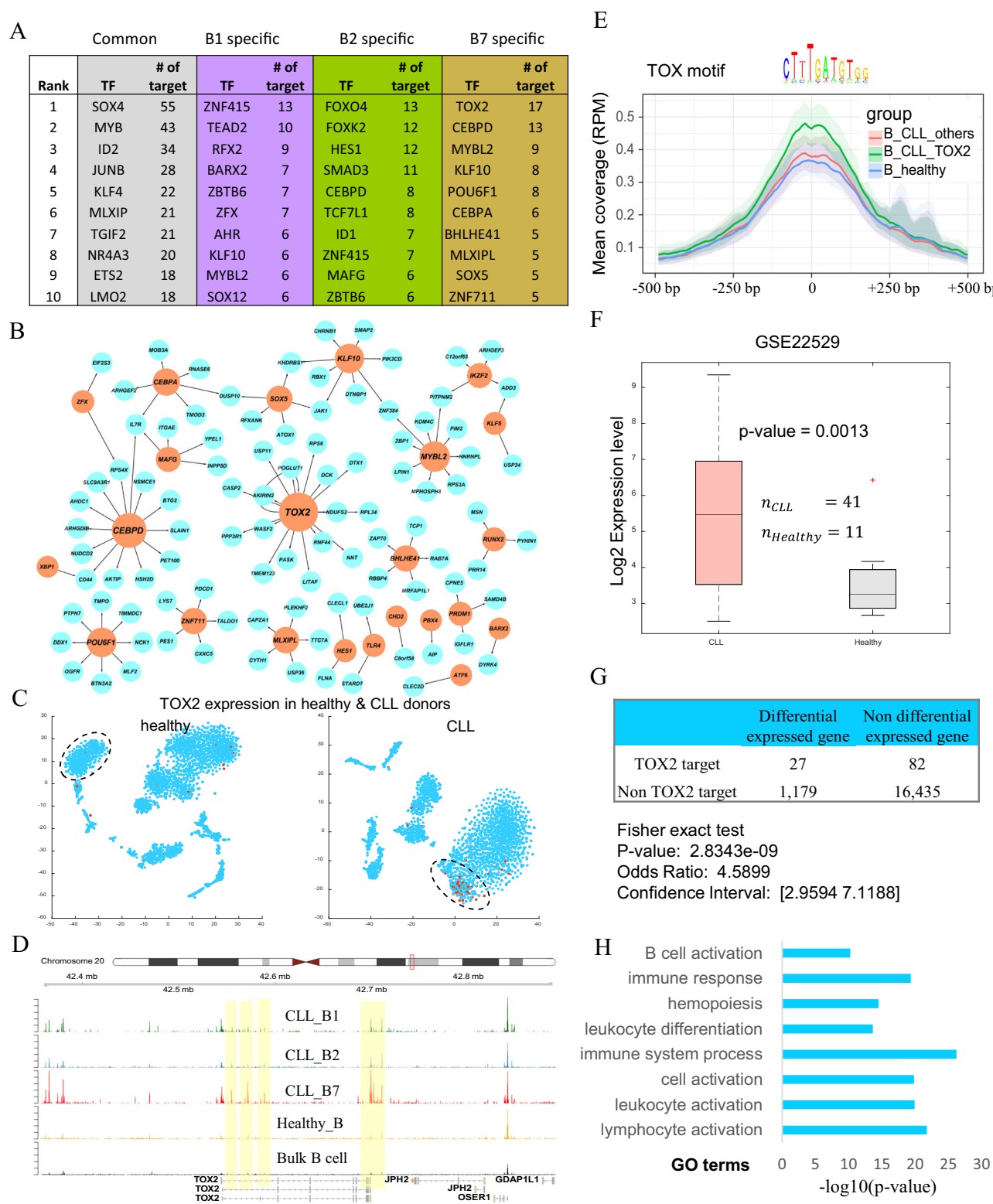

otherwise unknown. Cells were isolated and lysed following the Nuclei Isolation for Single Cell ATAC Sequencing Demonstrated Protocol (10x Genomics: CG000169) following guidance for primary cells. Sequencing libraries were generated following the Chromium Single Cell ATAC Reagent Kits User Guide (10x Genomics: CG000168). Briefly, nuclei were tagmented in bulk. Single nuclei were encapsulated in GEMs containing a gel bead with a unique cell barcode, and genomic material was amplified in a linear amplification reaction. GEMs were broken, and genomic material was amplified via PCR appending a sample index and Illumina sequencing handles (P5/P7). Libraries were sequenced on the Illumina NovaSeq 6000 System according to the manufacturer's protocols.

**Single-cell RNA.** BMMCs from the healthy donor and CLL patient were obtained from AllCells as described in the previous section. Cells were isolated following the Sample Preparation Demonstrated Protocol: Fresh Frozen Human Peripheral Blood Mononuclear Cells for Single Cell RNA Sequencing (10x Genomics: CG00039). Sequencing libraries were generated following the Single Cell 3′ Reagent Kits v2 User Guide (10x Genomics: CG00052). Briefly, cells were encapsulated in GEMs containing a gel bead with a unique cell barcode, and mRNA was reverse transcribed. GEMs were broken, and cDNA was amplified and fragmented. Ends were blunted, A-tailed, and adapter oligos were ligated. Samples were amplified via PCR appending a sample index and Illumina sequencing handles (P5/P7). Libraries

**Fig. 5 Identification of TOX2 as a tumor-specific regulator. A** Out-degree of TFs in differential networks. Common network consistent of edges that are differential in all the three networks. **B** Visualization of B7 specific differential network. The orange node represents TF, and the blue node represents non-TFs. The size of the node reflects the out-degree of the node. **C** t-SNE plot shows the expression of TOX2 in the healthy donor (left) and CLL donor (right). Red dots represent TOX2 expressed cells. **D** Track plot show cluster-specific peaks near TOX2. **E** Comparison of TOX motif binding REs' accessibility on different B cell subpopulations by Metagene plot. The central line represents the mean of accessibility, and the ribbon represents the minimum and maximum of average accessibility from 1000 times bootstrap analysis. **F** Comparison of TOX2 expression in B cell in independent CLL and healthy control study. The central mark indicates the median, the bottom and top edges of the box indicate lower and upper quartiles, the whiskers extend to the most extreme data points not considered outliers, and the outliers are plotted individually using the '+' symbol. We use a one-tailed two-sample $t$-test and the $p$-value without multiple testing adjustment is 0.0013. **G** Fisher's exact test shows TOX2 target genes are enriched in B7 specific genes (compared to healthy control). **H** GO enrichment analysis for TOX2 target genes.

were sequenced on the Illumina NovaSeq 6000 System according to the manufacturer's protocols.

**Statistical test for differential regulatory relations**. To test if the regulation of $i$-th TF to $j$-th TG is differential in two conditions, we propose the following method. This method takes the cell-level expression of TF and TG and subpopulation-level accessibility profile of RE from two conditions as input, and calculates a likelihood ratio and $p$-value for each pair of TF and TG in each subpopulation. Note that we only consider the TF−TG regulation edges in which TG is differentially expressed. Let TFRP be the regulatory potential of TF on TG, which is defined as follows

$$\text{TFRP}_{ijt} = \text{TF}_{it}\left(\sum_k B_{ik}\text{RE}_{kt}I_{kj}\right)$$

Where $t$ is the index of a single cell, $\text{TFRP}_{ijt}$ represents the regulatory potential of $i$-th TF on $j$-th TG in $t$-th cell; $\text{TF}_{it}$ represents the expression of $i$-th TF in $t$-th cell; $B_{ik}$ represents the motif binding strength of the $i$-th TF on the $k$-th RE; $\text{RE}_{kt}$ represent the subpopulation-level accessibility of $k$-th RE on the subpopulation matched to the subpopulation of $t$-th cell in scRNA-seq; $I_{kj}$ represents the interaction strength between $k$-th RE and $j$-th TG, which is learned from the PECA model based on diverse cellular contexts[31].

To identify differential edge for a differential expressed gene, we need to be able to detect both of the following two situations:

1. The TFRP is also differential in two conditions.
2. The association of TF and TG is differential in two conditions, e.g., strongly correlated in one condition and weakly correlated in the other condition.

As suggested in Fig. 2B, both of these can be detected by testing whether the conditional distributions P(TFRP|TG) are different in two conditions (note that testing P(TG|TFRP) is not informative because the distribution of TG is already shifted). We use a linear model for this conditional distribution:

$$\text{TFRP}_{ijt} = \beta_0 + \beta_1\text{TG}_{jt} + \varepsilon; \ \varepsilon \sim N(0, \sigma^2)$$

where $\text{TG}_{jt}$ represents the expression of $j$-th TG in $t$-th cell. We consider the likelihood ratio test for the null hypothesis H0 against the alternative H1 defined as follows. Let $(\beta_{c0}, \beta_{c1}, \sigma_c)$ be the parameters in $c$-th condition ($c = 1$ or 2), and $(\beta_0, \beta_1, \sigma)$ be the parameters learned by pooling all the cells from two conditions. Then the H0 and H1 are:

$$\text{H0:}(\beta_{10}, \beta_{11}, \sigma_1) = (\beta_{20}, \beta_{21}, \sigma_2)$$

$$\text{H1:}(\beta_{10}, \beta_{11}, \sigma_1) \neq (\beta_{20}, \beta_{21}, \sigma_2)$$

The likelihood under H0 is sharing the same parameter and $(\beta_0, \beta_1, \sigma)$ for both two conditions and the likelihood under H1 is using different parameters $(\beta_{10}, \beta_{11}, \sigma_1)$ and $(\beta_{20}, \beta_{21}, \sigma_2)$ for conditions 1 and 2 respectively. The likelihood under these two models are (here $f$ represents the probability density of the normal distribution):

$$(\text{M0}) L_0(\beta_0, \beta_1, \sigma) = \prod_t f(\text{TFRP}_{ijt}^{(1)}|\beta_0 + \beta_1\text{TG}_{jt}^{(1)}, \sigma^2)\prod_t f(\text{TFRP}_{ijt}^{(2)}|\beta_0 + \beta_1\text{TG}_{jt}^{(2)}, \sigma^2)$$

$$(\text{M1}) L_1(\beta_{10}, \beta_{11}, \sigma_1, \beta_{20}, \beta_{21}, \sigma_2) = \prod_t f(\text{TFRP}_{ijt}^{(1)}|\beta_{10} + \beta_{11}\text{TG}_{jt}^{(1)}, \sigma_1^2)\prod_t f(\text{TFRP}_{ijt}^{(2)}|\beta_{20} + \beta_{21}\text{TG}_{jt}^{(2)}, \sigma_2^2)$$

The likelihood ratio statistics for $i$-th TF and $j$-th TG pair is:

$$\text{LR}(\text{TF}_i, \text{TG}_j) = -2\ln\left[\frac{\max\limits_{(\beta_0, \beta_1, \sigma)} L_0(\beta_0, \beta_1, \sigma)}{\max\limits_{(\beta_{10}, \beta_{11}, \sigma_1, \beta_{20}, \beta_{21}, \sigma_2)} L_1(\beta_{10}, \beta_{11}, \sigma_1, \beta_{20}, \beta_{21}, \sigma_2)}\right] \sim \Gamma(\kappa, \theta)$$

In theory, under standard conditions, the likelihood ratio should have a chi-square distribution (with df = 3) as its null distribution. Figure 2F shows that the chi-square is no longer an adequate approximation for the null distribution.

Instead, the null distribution is well fitted by a two-parameters Gamma distribution. The question, then, is how to infer the gamma parameters in a real application when we do not know the null (i.e. non-differential) TF−TG relations. Here we assume that most of the TF−TG pairs are non-differential so that the lower quantiles of the empirical distribution of the LR statistics should be approximately the same as those in the true null distribution. Accordingly, we infer the parameter of the Gamma by fitting the lower quantiles of the empirical distribution. Specifically, let $q(i; \alpha, \beta)$ be the $i*(0.05)$-th quantile of the Gamma($\alpha$, $\beta$) distribution, and $t(i)$ be the corresponding quantile of the empirical distribution of LR statistics. We infer $\alpha$ and $\beta$ by minimizing

$$G(\alpha, \beta) = \sum_{i=1}^{10}(q(i; \alpha, \beta) - t(i))^2$$

**Coupled clustering**. Here we introduce a modification of our previous Coupled Clustering model[9]. Given scATAC-seq data count matrix $O$ ($p_1$ regions by $n_1$ cells matrix) and scRNA-seq transcripts per kilobase million (TPM) data matrix $E$ ($p_2$ genes by $n_2$ cells matrix), we do coupled clustering based on the CoupledNMF optimization model. We do quantile normalization and log transformation (log2 on expression and log10 on accessibility) on two data matrices. A "soft" clustering of scATAC-seq can be obtained from a nonnegative matrix factorization $O = W_1H_1$ as follows: the $i$th column of $W_1$ gives the mean vector for the $i$th cluster of cells, while the $j$th column of $H_1$ gives the assignment weights of the $j$th cell to the different clusters. Similarly, the clustering of the scRNA-seq can be obtained from the factorization $E = W_2H_2$. To simplify the original CoupledNMF model, we move one term from the objective function to constrains so that the number of tuning parameters is decreased from 3 to 2. Here is the modified constrained optimization version of the CoupledNMF model:

$$\min_{W_1, H_1, W_2, H_2} \frac{1}{2}||O - W_1H_1||_F^2 + \frac{\lambda_1}{2}||E - W_2H_2||_F^2 - \lambda_2\text{tr}(W_2^T A W_1) \quad (1)$$

s.t. $\sum_{i=1}^{i=p_1}\left(w_{ij}^1\right)^2 = 1, j = 1,2,\ldots K;$

$\sum_{i=1}^{i=p_2}\left(w_{ij}^2\right)^2 = 1, j = 1,2,\ldots K;$

$W_1, H_1, W_2, H_2 \geq 0;$

There are two tuning parameters: $\lambda_1$ and $\lambda_2$. Where $A$ is the predefined coupling matrix by considering peak to target gene distance and correlation of peak accessibility and target gene expression across diverse cellular contexts[9].

$$A_{ij} = R_{ij}^+ \times e^{-d_{ij}/d_0}$$

Where $R_{ij}^+$ represents the correlation of $i$-th gene expression and $j$-th peak accessibility across diverse cellular contexts (negative values are set to zeros); $d_{ij}$ represents the distance between RE and TG; the base parameter $d_0$ reflects the scale over which the coupling coefficient decreases with distance (default value is 40 kb).

**Algorithm of CoupledNMF**. We design an algorithm for the constrained optimization version of the CoupledNMF model. Update of variables are as follows:

$$w_{ij}^1 \leftarrow w_{ij}^1 \frac{\left(OH_1^T + \lambda_2 A^T W_2 + W_1\text{diag}\left(W_1^T\left(W_1H_1H_1^T\right)\right)\right)_{ij}}{\left(W_1H_1H_1^T + W_1\text{diag}\left(W_1^T\left(OH_1^T + \lambda_2 A^T W_2\right)\right)\right)_{ij}}$$

$$w_{ij}^2 \leftarrow w_{ij}^2 \frac{\left(EH_2^T + \frac{\lambda_2}{\lambda_1}AW_1 + W_2\text{diag}\left(W_2^T\left(W_2H_2H_2^T\right)\right)\right)_{ij}}{\left(W_2H_2H_2^T + W_2\text{diag}\left(W_2^T\left(EH_2^T + \frac{\lambda_2}{\lambda_1}AW_1\right)\right)\right)_{ij}}$$

$$h_{ij}^1 \leftarrow h_{ij}^1 \frac{\left(W_1^T O\right)_{ij}}{\left(W_1^T W_1 H_1\right)_{ij}}$$

$$h_{ij}^2 \leftarrow h_{ij}^2 \frac{(W_2^T E)_{ij}}{(W_2^T W_2 H_2)_{ij}}$$

Here diag(X) represents a diagonal matrix that has the same main diagonal with matrix X. We show that the update algorithm correctly solves this optimization problem. To simplify this, we just talk about the update of the variable $W_1$. We introduce the Lagrangian multipliers $\nu$(a diagonal matrix with element $[\nu_1, \nu_2, \dots \nu_K]$). The Lagrangian multiplier $\nu_j$ corresponds to the constraint $\sum_{i=1}^{i=p_1} (w_{ij}^1)^2 = 1$. We minimize the Lagrangian function

$$L(W_1) = \frac{1}{2}||O - W_1 H_1||_F^2 - \lambda_2 tr(W_2^T A W_1) - \sum_{(j=1)}^{K} \nu j \left( \sum_{(i=1)}^{(i=p_1)} (w_j^1)^2 - 1 \right)$$

The gradient is:

$$\frac{\partial L}{\partial W_1} = W_1 H_1 H_1^T - O H_1^T - \lambda_2 A^T W_2 - 2 W_1 \nu.$$

The KKT condition for the nonnegativity of $w_{ij}^1$ gives

$$(W_1 H_1 H_1^T - O H_1^T - \lambda_2 A^T W_2 - 2 W_1 \nu)_{ij} w_{ij}^1 = 0 \quad (2)$$

By combining this equation with the constrains $\sum_{i=1}^{i=p_1} (w_{ij}^1)^2 = 1$, we get

$$2\nu = diag(W_1^T(W_1 H_1 H_1^T - O H_1^T - \lambda_2 A^T W_2)). \quad (3)$$

By combining Eqs. (2) and (3), we get

$$\left( W_1 H_1 H_1^T - O H_1^T - \lambda_2 A^T W_2 - W_1 diag\left( W_1^T(W_1 H_1 H_1^T - O H_1^T - \lambda_2 A^T W_2) \right) \right)_{ij} w_{ij}^1 = 0 \quad (4)$$

The update rule of $W_1$ satisfies this Eq. (4).

**Parameters selection of coupled clustering**. Local minima of standard NMF should satisfy $O H_1^T \approx W_1 H_1 H_1^T$, and $E H_2^T \approx W_2 H_2 H_2^T$. We hope the solution of our optimization problem in Eq. (1) is close to one of the local minima of the standard NMF. We introduce two interpretable parameters $\alpha$ and $\beta$

$$\alpha\beta = \frac{mean\left( \frac{\lambda_2}{2} A^T W_{20} \right)}{mean\left( O H_{10}^T \right)}$$

$$\alpha(1 - \beta) = \frac{mean\left( \frac{\lambda_2}{2\lambda_1} A W_{10} \right)}{mean\left( E H_{20}^T \right)}$$

where $W_{10}, H_{10}, W_{20}, H_{20}$ are the solution of standard NMF; parameter $\alpha$ is nonnegative and reflects the ratio of coupling term and factorization term; parameter $\beta$ ranges from 0 to 1 and balances the weight of scATAC-seq and scRNA-seq. The default value of parameter $\alpha$ is 0.1%, which means in each iteration, the effect of the coupling term on matrix factorization is about 0.1%. The default value of parameter $\beta$ is 0.5, which means we think scATAC-seq and scRNA-seq are equally important.

To stabilize the results, we suggest the following strategy. We start with a random initial $W_1, H_1, W_2, H_2$ and a bigger $\alpha$ parameter to get the initial solutions. Then we decrease the $\alpha$ parameter gradually and take the former step solution as the initial. We suggest the $\alpha$ parameter series is [10000, 1000, 100, 10, 1, 0.1, 0.01, 0.001].

**Construction of surrogate ground truth**. We use publicly available RNA-seq and ATAC-seq data on 13 distinct cellular populations from the human hematopoietic hierarchy isolated via FACS to construct ground truth labels for the single-cell data[32].

RNA-seq: We define a specific gene set for each of the cell types by choosing the top 1000 specifically expressed genes (expression/median expression across 13 cell types). For a given single cell and a cell type, we count the number of expressed (read count > 0) specific genes in the single-cell data. We assign the single cell to the cell type which has the maximum number of specific genes expressed.

ATAC-seq: For each of the cell types' ATAC-seq data, we get a peak set from MACS2. For the i-th single cell and j-th cell type, we count the number of open (read count > 0) peaks of the cell type in the single-cell data (noted as $c_{ij}$). This count matrix C is biased because the number of peaks in different cell types is different. To remove the bias, we do quantile normalization to matrix C to make the different cell types comparable. We label the i-th single cell to the cell type which has the maximum normalized $c_{ij}$ value.

Because the number of progenitor cells is too small and difficult to be identified, we merge all the progenitor cells into one label. So we get seven labels in total (B cell, CD4 T cell, CD8 T cell, Erythroid, Monocyte, natural killer, and progenitors).

**Evaluation of clustering performance**. To evaluate the clustering performance, we link the clusters to the seven true labels by bipartite graph maximum weighted matching. Given i-th cluster and j-th true label, we count the number of overlapping cells (noted as $S_{ij}$) in the i-th cluster and cells have the j-th true label. Based on the S matrix, we use the Hungarian algorithm to do a one-to-one matching between the clusters and the true labels. This matching will maximize the accuracy

which is defined as the fraction of correctly predicted cells.

$$accuracy_j = \frac{S_{ij}}{\sum_k S_{kj}}$$

Where i is the best matching cluster for the j-th true label. This accuracy metric is sensitive to minor population detection. If one minor population is merged into a major population, it will have very low accuracy. Except for this accuracy-based evaluation, we also use normalized mutual information to evaluate the clustering.

**Joint embedding of scRNA-seq and scATAC-seq**. Coupled clustering model will do linear transformations for scATAC-seq and scRNA-seq to get $H_1$ and $H_2$ matrices, which are in a common K dimensions space and comparable to each other. Based on the normalized (column square sum to one) $H_1$ and $H_2$ matrices, we use tSNE to reduce them from K dimensions to two dimensions.

**Cluster specific features**. To get the cluster-specific peaks, we treat the data as binary and use the Fisher exact test to calculate the p-values and fold changes. For scRNA-seq data, we use a one-tailed two-sample t-test to calculate the p-value and fold changes for genes. To calculate fold change, we add a pseudocount 0.01 on the denominator. We combine the p-values and fold changes to define a cluster specificity score by $-\log_{10}(pvalue) \times$ FoldChange.

**Linking of subpopulations across conditions**. To perform comparative regulatory analysis, the subpopulations across two populations are required to be linked. Considering the batch effect between different populations, we use a two-step strategy of clustering and linking instead of joint clustering across two populations using two types of single-cell data, which is a more challenging task. To map the subpopulations across two donors, we define a mapping score to evaluate the similarity of two subpopulations from two donors based on expression and accessibility profile. The mapping score consists of two parts, scRNA-seq based mapping score and scATAC-seq based mapping score. The scRNA-seq based mapping score of i-th cluster from population 1 and j-th cluster from population 2 is defined as follows

$$\text{Mapping score}_{ij}^{rna} = r_{ij}^{rna} - \frac{\left( \sum_i r_{ij}^{rna} \right) \left( \sum_j r_{ij}^{rna} \right)}{\sum_i \sum_j r_{ij}^{rna}}$$

Here $r_{ij}^{rna}$ represent the Pearson correlation coefficient (PCC) of the two cluster-specific expression profiles. Intuitively, the mapping score presents the observed PCC minus the expected PCC. Similarly, we define the mapping score for scATAC-seq data. We map i-th cluster from population 1 to j-th cluster from population 2 if both Mapping score$_{ij}^{rna}$ and Mapping score$_{ij}^{atac}$ have positive values. The confidence of the cluster mapping is reflected by the mapping score, which is defined as the sum of Mapping score$_{ij}^{rna}$ and Mapping score$_{ij}^{atac}$.

**Differential regulatory network data simulation**. We simulate a K TF (k-th TF), N gene (j-th TG), and M RE (i-th RE) network on H cells (h-th cell) from two conditions (c-th condition). We simulate the target gene expression by the formulation provided by the PECA model[31]. The expression of j-th TG on the h-th cell is simulated as

$$TG_{jh} = \nu_j + \sum_k \gamma_{kj} \left( \sum_i B_{ik} \beta_{ij} RE_{ic} \right) TF_{kh} + \varepsilon; \varepsilon \sim N\left( 0, (0.5\nu_j)^2 \right)$$

Where the $\nu_j$ is the baseline expression of j-th TG which is generated from a Gamma distribution (details see Supplementary Fig. S16). To simulate TG expression by PECA formulation, we require TF expression, RE accessibility, TF −RE binding affinity, RE−TG association, and TF−TG regulatory structure. We use a normal distribution to generate the TF expression and RE accessibility for different cells, where the mean of TF expression and RE accessibility is generated from a gamma distribution. After generating the "real" expression, we simulate the drop-out and observe the gene expression. We randomly select some TF and RE, and add some value to them to simulate the differential TF and RE. TF−RE motif binding affinity and RE−TG connections are generated from a Bernoulli distribution. TF−TG connection should be very sparse so it is simulated as a mixture of a normal distribution with zero mean and a point mass at zero. To simulate differential regulatory structure, we randomly select some non-zero TF−TG connections and set their values to zeros. Note that the model used to generate the simulation data and the model to be evaluated (sc-compReg) are totally different. The formulation of simulated data is based on the PECA model, and the expression of target genes is determined by the combined effect of multiple TFs. But the model in sc-compReg just focused on the pairwise relations not considering the joint effect of multiple TFs. Thus, this simulation provides a fair comparison of sc-compReg with other alternative methods. See the detailed formulation as follows:
Simulation of TF expression:
TF mean expression $\mu \sim$ gamma(2, 2)
TF expression $TF_k \sim N(\mu_k, \mu_k^2)$
Differential TF $TF_{kh} = TF_{kh} + r_{kh}\mu_k$; $r_{kh} \sim U(0.1, 1)$

Simulation of RE openness profile:

RE means openness $\nu \sim \text{gamma}(1,2)$

RE openness $\text{RE}_i \sim N(\nu_i, (0.01\nu_i)^2)$

Differential RE $\text{RE}_{ic} = \text{RE}_i + r_{ic}\nu_i$; $r_{ic} \sim U(0.1, 1)$

Simulation of the regulatory network:

TF motif binding $B_{ik} \sim \text{Bernoulli}(10/K)$

RE−TG connection $\beta_{ij} \sim \text{Bernoulli}(10/M)$

TF−TG network structure $\gamma_{kj} \sim 0.1N(0,1) + 0.9\delta_0$

Differential network structure $\gamma_{kj}^{(c)} = \omega_{kj}^{(c)}\gamma_{kj}$; $\omega_{kj}^{(c)} \sim \text{Bernoulli}(0.01)$

Simulation of target gene expression:

TG mean expression $\nu \sim \text{gamma}(2,2)$

TG expression $\text{TG}_{jh} = \nu_j + \sum_k \gamma_{kj}(\sum_i B_{ik}\beta_{ij}\text{RE}_{ic})\text{TF}_{kh} + \varepsilon$; $\varepsilon \sim N(0, (0.5\nu_j)^2)$

Simulation of drop-out:

For both TF and TG, their probability of drop-out is $\frac{1}{1+e^{-\kappa\log(E)-\lambda}}$; where $E$ represents the expression level, we choose parameters $\kappa$ as 0.1 and the $\lambda$ as 0.5. For each expression, we generate a random variable from the standard uniform distribution $U(0,1)$. The expression level will be dropped to zero if this generated number is less than $\frac{1}{1+e^{-\kappa\log(E)-\lambda}}$. A gene with a lower expression level is more likely to be dropped out.

**Copy number analysis**. We use R package CopyKAT to infer the genomic copy number from the scRNA-seq data of B1, B2, B7, and healthy B cells. Based on the inferred copy number, the CopyKAT classifies cells into two groups named aneuploid and diploid. We do Fisher's exact test between aneuploid/diploid groups and cell clusters of B1, B2, and B7 respectively (Supplementary Fig. S10A). For each window from CopyKAT, we compare the copy number distribution of B1, B2, and B7 with the healthy B cell copy number distribution by a two-sample t-test. We adjust the p-values by Bonferroni correction and treat the FDR less than 0.05 is significant.

**TOX motif**. We download both wild type and TOX knock-out ATAC-seq data on CD8 T cell from ref. [29] and infer TOX-dependent open regions (higher openness in wild type) by differential openness analysis (one-tailed two-sample t-test, p-value < 0.05 and Fold > 1.5). On the 9,597 TOX-dependent regions, we do de novo motif enrichment analysis and find the TOX motif (Supplementary Fig. S13). The new identified TOX motif has been included in the database of PECA software https://github.com/SUwonglab/PECA.

**Reporting summary**. Further information on research design is available in the Nature Research Reporting Summary linked to this article.

## Data availability

The healthy and CLL donors' scRNA-seq and scATAC-seq data generated in this study have been submitted to the NCBI Gene Expression Omnibus (GEO; https://www.ncbi.nlm.nih.gov/geo/) under accession number GSE159417. The FACS sorted bulk data is downloaded from the NCBI Gene Expression Omnibus (GEO; https://www.ncbi.nlm.nih.gov/geo/) under accession number GSE75384.

## Code availability

R version of the sc-compReg software[33] is available at https://github.com/SUwonglab/sc-compReg. A Matlab version of the sc-compReg and processed data are also available in https://github.com/durenzn/sc-compReg_matlab.

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

## Acknowledgements

This work was partially supported by NIH grants R01 HG010359 and P50 HG007735.

## Author contributions

W.H.W. and J.A. conceived the project. Z.D. designed the analytical approach, performed data analysis with the help of J.X. and M.L. Z.D. and W.L. wrote the software. C.N., Y.Y. and F.M. performed all biological experiments. Z.D., W.H.W., J.A. and P.S. wrote, revised, and contributed to the final paper.

## Competing interests

J.A., P.S., F.M., C.N. and Y.Y. are employees of 10x Genomics. The remaining authors declare no competing interests.

## Ethics approval

Bone Marrow and/or Peripheral Blood Collection From Disease-Specific Donors for the Research Market (7000-SOP-078), and Bone Marrow Collection from Healthy Donors for the Research Market (7000-SOP-046) were approved by Alpha IRB. Informed consent was obtained from the donors, and all ethical regulations have been followed.

## Additional information

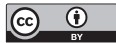

