## [Peer Review File · Nature Communications]

Reviewers' Comments:

Reviewer #1:

Remarks to the Author:

Wong and colleagues have described a novel statistical framework, *sc-compReg*, that integrates scRNA-seq and scATAC-seq data generated from 10X platform and identifies regulatory networks that drive the transcriptional phenotypes of one sample compared to the other sample.

Integration of scRNA-seq data and scATAC-seq data is an important and timely problem in the field. This software will allow an integrated analysis of data generated from the two platforms, allowing the identification of true molecular drivers. The statistical and mathematical framework for the *sc-compReg* appears robust based on the described method and simulation result presented, although I do not have full expertise in validating their mathematical methodology (Method and Figure 1).

The authors used a bone marrow sample from CLL patient as an example of the use of this platform to identify CLL-specific gene regulatory network. The pipeline identified 3 B cell subpopulations (B1, B2, and B7) in CLL bone marrow, of which B7 was specific to CLL sample. Then the authors identified TOX2 as a central regulatory driver of the B7 subpopulation.

- CLL cells demonstrate unique immunophenotype (CD5+CD23+CD19+). This can be easily assessed from scRNA-seq data. The obvious question is whether B1, B2, and/or B7 cells show this unique phenotype or not. Since CLL bone marrow can contain normal B-cells, it is important to identify the malignant population. Simply linking B-cells between CLL donor and healthy donor is not sufficient for comparing malignant vs. normal B-cells.

-There is also no information about the patient demographics, immunophenotype, genetic data on this CLL donor. CLL cells can show increased clonal heterogeneity, and the question is whether the subpopulations that the authors identified in CLL B cells have anything to do with the genetic heterogeneity (genetic subclonality) or even immunophenotypic heterogeneity.

- I am not so sure about the purpose of Figure 4D. They compare the frequency of TP53 expressed cells among B1, B2, B7, and healthy donor's B cells. But what does this mean? There is no interpretation done. Also, the difference appears very small. 8.4% vs. 3-5%. It is not clear how this information is helping the authors argument.

- The paper does not discuss or compare the performance with another method such as MAESTRO. At least this needs to be added as a discussion.

- The figures are shown in dissociated orders. It is recommended that figures are shown in order of how it is discussed in the text.

- Figure 4F and 4H has no unit for the x and y-axis.

Reviewer #2:

Remarks to the Author:

This manuscript presents *sc-compReg*, a method for identifying differential regulatory interactions between two cell populations based on scRNA-seq and scATAC-seq data. The problems of regulatory network inference and differential network analysis are well established with clear motivations in systems biology. This work is relevant to the increasing availability of single-cell multi-omics data sets combining scRNA-seq with scATAC-seq, for example, that present the opportunity to improve on existing algorithms. The authors also contribute an implementation for the joint embedding and clustering of cells based on the integration of gene expression and chromatin accessibility profiles, which is a step common to several downstream analyses of multi-omics data. There are interesting

results presented for the identification of TOX2 as a tumor-specific regulator in chronic lymphocytic leukemia that are supported by further analysis and related literature.

MAJOR COMMENTS

While the authors introduce broad categories of relevant literature in Figure 1A with a few citations, there is not any citation or discussion of existing research that is most relevant to this manuscript, i.e., in the category of the comparative analysis of gene regulatory networks (Figure 1A, lower right). There are existing algorithms for gene regulatory network inference based on various forms of gene expression data, including the combination of scATAC-seq and scRNA-seq (e.g., Building gene regulatory networks from scATAC-seq and scRNA-seq using Linked Self Organizing Maps, <https://journals.plos.org/ploscompbiol/article?id=10.1371/journal.pcbi.1006555>). There is also substantial literature regarding the application of differential network analysis for the identification of (differential) interactions in gene regulatory networks (e.g., Comparative assessment of differential network analysis methods. <https://academic.oup.com/bib/article/18/5/837/2562785> and DNF: A differential network flow method to identify rewiring drivers for gene regulatory networks, <https://www.sciencedirect.com/science/article/pii/S0925231220308638>). The authors must provide a detailed discussion of the literature and provide a convincing argument that their research represents an advance over existing methods. There is certainly the potential for novel contributions in the proposed statistical method and the integration of scRNA-seq with scATAC-seq data, but it is not clear that these improve the performance of identifying differential regulatory interactions in comparison to methods that already exist in the literature.

In this context, the validation of the proposed likelihood test for the identification of differential regulatory interactions is not sufficient. The results shown in Figure 2D and 2E demonstrate that the AUC for predictions made by the proposed method increases as additional accessibility information is integrated with gene expression to determine a transcription factor's regulatory potential. These are the primary results that establish the performance of this method before its application to chronic lymphocytic leukemia. However, all of these results are based on simulated data, and there is no comparison between sc-compReg and any other differential network analysis algorithms.

* There needs to be further discussion in the differential regulatory network data simulation section of the methods in order to justify why the simulated gene expression, regulatory element accessibility, and regulatory networks are representative of biological data.

* Additionally, the performance of sc-compReg has to be measured against some form of literature curated or experimentally validated gene regulatory networks, using experimental scRNA-seq and scATAC-seq data.

* While published algorithms may not directly identify differential regulatory interactions in scRNA-seq with scATAC-seq data, regulatory networks output by existing GRN inference algorithms could be used as input to existing differential network analysis methods in order to compare their predictions with those made by sc-compReg and with ground truth networks. Benchmarking results could be used to identify the best performing regulatory network inference (e.g. Benchmarking algorithms for gene regulatory network inference from single-cell transcriptomic data, <https://www.nature.com/articles/s41592-019-0690-6>) and differential network analysis (e.g., Comparative assessment of differential network analysis methods, <https://academic.oup.com/bib/article/18/5/837/2562785>) algorithms, available data sets and ground truth regulatory networks, and evaluation metrics for the validation of sc-compReg performance.

* In a comparison of sc-compReg with a combination of GRN inference and differential network analysis algorithms that do not integrate scATAC-seq with scRNA-seq data, sc-compReg results could be shown both with and without incorporating accessibility information in the calculation of transcription factor regulatory potential, as in the current Figure 2, in order to demonstrate the degree to which an improvement in performance results from the proposed likelihood ratio calculation or the inclusion of accessibility information from scATAC-seq data.

The description of the rigorous validation methodology and the claim that sc-compReg is well-validated using data collected from bone marrow mononuclear cells is misleading. This strategy is only applied to the joint embedding and clustering approach, and is not used to validate the proposed likelihood ratio statistic for differential regulation detection, which is the primary novel contribution of this paper.

- * It is necessary to be more precise in the definitions of the input, output, and variables referenced in the statistical test for differential regulatory relations section of Methods.
- * There isn't an explicit definition for the values of the input variables TFit and TGjt.
- * It isn't clear from the definition of the output variable LR that a likelihood ratio statistic is computed for each TFRP and TG pair.
- * The parameters T1 and T2 in the LR calculation are not defined.
- * There are subtle changes in the notation of related variables that should be explained, such as the meaning of the difference between X, X_{ht}, and X_t and the difference between β_{c0} , β_{c1} and β_0 , β_1 . For example, in the linear model, I presume that X is a matrix of expression values for all genes across all cells. Is that correct? Then what is X_t in (M0) and (M1)?
- * The notation might be simplified by replacing the subscript h, which denotes the h-th regulation between the i-th TF to j-th TG, with the subscript ij.
- * Additionally, the notation shown in the related definition of TFRP and LR in Figure 2A and Figure 2C should be consistent with the methods, and the caption of Figure 2 should either reference the related methods or separately define the variables shown in the figure.
- * Please provide rationale for the formulae for the likelihoods in (M0) and (M1). The rationales are clear after viewing the formulae carefully but a sentence or two of explanation will be helpful for the reader. It does not seem necessary to number all the cells from 1 to T1+T2. It is probably easier for the reader if the cells in the two conditions are numbered individually from 1 to T1 and from 1 to T2. That way both (M0) and (M1) will have two products, making the correspondence and the differences between them more apparent to the reader.

The proposed validation methodology is convincing for the evaluation of joint embedding and clustering, although it isn't clear based on Figure 3 and Supplementary Note 3 that CoupledNMF significantly outperforms the other algorithms. While the accuracy for cell types noted in Supplementary Note 3 is higher for CoupledNMF, the results shown in Supplementary Figure S7 indicate there are other cell types in which CellRanger and Seurat have higher accuracy. A statistical comparison such as a two sample Kolmogorov-Smirnov test could be used to report a p-value demonstrating that the distribution of accuracy achieved by CoupledNMF is significantly higher than the distribution of accuracy achieved by CellRanger and Seurat. Similarly, the claim that cells of the same ground truth label are placed near each other in the joint embedding would be more convincing if it were supported by a quantitative measurement of the distance between cells with the same label in addition to referencing the plot in Figure 3D.

MINOR COMMENTS

The Coupled clustering and Algorithm of CoupledNMF sections of the methods are closely related to the prior publication "Integrative analysis of single-cell genomics data by coupled nonnegative matrix factorizations," <https://www.pnas.org/content/pnas/115/30/7723.full.pdf>. The definition of TFRP in the Statistical test for differential regulatory relations section is closely related to the prior publication "Modeling gene regulation from paired expression and chromatin accessibility data," <https://www.pnas.org/content/pnas/114/25/E4914.full.pdf>. As written, the Methods and the Result present the impression that these methods are proposed in this manuscript. The authors should change the language to make it clear that these two methods have already been published. For example, they could describe newly proposed and existing methods under separate headers, and add citations for equations referenced from existing publications in order to highlight the novel contributions of this paper.

The order of the sections in Methods isn't consistent with the order in which the results are presented. It will be more clear to change the order of the methods, specifically, to describe the new statistical method before presenting the coupled clustering algorithm. This change will also highlight the new statistical method.

It is difficult to distinguish between the solid and dashed lines in Figure 2D, especially when the lines overlap with each other or the axes of the plot. It would improve the readability of Figure 2D to use different colors for each line, and either increase the size of the plots or change the scale of the axes to further separate the two lines from each other and the axes.

References to the upper left, upper right, lower left, and lower right cells of Figure 1A can be further clarified by adding alphanumeric labels for each category (e.g. Figure 1A, i ; Figure 1A, ii)

The caption of Figure 4B should contain a reference to the sub-section of Methods describing the linking of subpopulations across conditions. This change will clarify the meaning of PCC and mapping score. An alternative is to include those definitions in the caption.

Reviewer #3:

Remarks to the Author:

Duren et al. presents a new computational method sc-compReg. This method aims to use single cell chromatin accessibility and gene expression experiments, performed across two conditions, to infer how regulation by transcription factors differs between the two conditions. This "differential regulation" is only meaningful in the context of the same cell type across different conditions. Therefore, differential regulation estimation requires as an input a set of annotated cells, linked across experimental conditions. The authors provide software for this purpose.

The analysis their method enables would be of interest within the research community and their example applying it to a chronic lymphocytic leukemia adequately demonstrates its utility. There are some details of the methodology and presentation that could be improved (see below), but the main limitation of this method is the choice to implement it in a closed scripting language. This choice has a number of consequences. Firstly, it compels Duren et al to reinvent a series of well established routines for clustering and integration of annotation across experiments. An attempt is made to demonstrate that the methods created to perform these tasks are comparable to other established methods, which at best demonstrate that their methods are "good enough". They certainly do not advance the state of the art in any way.

Probably the worst consequence of providing an implementation requiring proprietary software to run is that it will limit its uptake by the community. By and large, a computational method is only as useful as the software that implements it. Were their method implemented in an open language such as python or R, I could see it becoming the standard tool for differential regulation analysis of single cell RNA/ATAC data. As it stands, it is unlikely to be used outside those with an extremely strong need to do this analysis and it will likely be abandoned as soon as an open implementation becomes available.

That said, there is nothing fundamentally flawed about this paper or the method it describes that I can see. It will enable researcher to perform a specific type of analysis that will likely be of use to some. It is just a great shame that it's use and uptake is likely to be severely limited by the details of the software implementation.

Major criticisms:

1) One of the major limitations of the proposed method is its implementation in a proprietary software language. As I do not have a matlab license I could not test the sc-compReg and this is likely to

greatly limit the appeal and utility of their method.

2) I am not convinced that a new clustering and annotation method is needed. There are already a number of well established methods and software packages for clustering of single cell expression/ATAC data (e.g. scanpy/Seurat). I suspect the real reason this wheel has been reinvented here is because of point 1, that their choice of scripting language meant that they can not easily take advantage of the existing ecosystem of analysis tools in R/python.

3) Similarity, there are many methods to identify similar populations of cells across two experiments.

4) The authors do provide a comparison of their clustering routines to other methods (Fig 3) which I would describe as only mildly convincing. This comparison relies on a torturous definition of ground truth which itself in effect applies label transfer from FACs sorted bulk data to single cell data. It would be far better to rely on biologically motivated manual annotation of existing data (such as PBMCs) as the ground truth. Leaving these concerns to one side, I would not say that the new "CoupledNMF" method performed better than existing approaches. All in all, the comparison is probably enough to demonstrate that the approach in the manuscript is "good enough" in the absence of alternatives, but not to justify its use over established approaches.

Other comments:

1) The figures are very heavy on text/formula that cannot be understood with just the figure plus legend alone (e.g. Fig 2C). Such details would be better included in the text/methods where they should be explained in full.

2) It is not clear to me that H_0 is nested within H_1 , as is required for the likelihood ratio statistic to follow a chi-squared distribution. Could the fitting procedure on X be reformulated in terms of the difference between the two conditions? The null would then become $\beta=0$, with the alternative $\beta \neq 0$.

3) The change in TP53 expression level between populations is not particularly striking and while I'm sure it is statistically significant this is insufficient to decide which populations are tumour. Does the sample being considered possess somatic mutations in TP53?

Reviewer #1 (Expertise: The use of sc technologies in blood cancers):

Wong and colleagues have described a novel statistical framework, sc-compReg, that integrates scRNA-seq and scATAC-seq data generated from 10X platform and identifies regulatory networks that drive the transcriptional phenotypes of one sample compared to the other sample. Integration of scRNA-seq data and scATAC-seq data is an important and timely problem in the field. This software will allow an integrated analysis of data generated from the two platforms, allowing the identification of true molecular drivers. The statistical and mathematical framework for the sc-compReg appears robust based on the described method and simulation result presented, although I do not have full expertise in validating their mathematical methodology (Method and Figure 1).

Response: Thank you for your positive comments. We have implemented the new method in R and C++ based on the reviewers' and editor's comments, so it took longer than expected. We apologize for keeping you waiting.

Comment 1: The authors used a bone marrow sample from CLL patient as an example of the use of this platform to identify CLL-specific gene regulatory network. The pipeline identified 3 B cell subpopulations (B1, B2, and B7) in CLL bone marrow, of which B7 was specific to CLL sample. Then the authors identified TOX2 as a central regulatory driver of the B7 subpopulation.

- CLL cells demonstrate unique immunophenotype (CD5+CD23+CD19+). This can be easily assessed from scRNA-seq data. The obvious question is whether B1, B2, and/or B7 cells show this unique phenotype or not. Since CLL bone marrow can contain normal B-cells, it is important to identify the malignant population. Simply linking B-cells between CLL donor and healthy donor is not sufficient for comparing malignant vs. normal B-cells.

Response: Thank you for your comments. In light of your comments, we have compared the immunophenotype of those B cell subpopulations. The following description is added to the main text.

"We further compare those B cell subpopulations based on immunophenotype. Based on previous studies²⁰, well known surface marker of B cell CLL includes CD5, CD23, CD19, CD25, CD69, and CD71. In our data, CD23, CD19, and CD69 are expressed greater than 5% of the cells. Low expression of the other three markers may be because of the drop-out of single cell measurement. We systematically compare the pattern of markers CD23, CD19, and CD69 in the four different B cell subpopulations. Specifically, for each of the subpopulations (B1, B2, B7 and normal B cells), we first binarize the data as expressed or not expressed, and then calculate the distribution of cells among the 8 types expression patterns (i.e. CD23+CD19+CD69+, CD23+CD19+CD69-, CD23+CD19-CD69-, etc.). The degree of immunophenotype dissimilarity between two subpopulations can be quantified by the Hellinger distance between the two corresponding distributions. (Fig. 4D). It is seen that, among the three B cell subpopulations in CLL, B1 shows the highest similarity to normal B cells and B7 shows the least similarity to normal B cells. This is consistent with the global pattern from scRNA-seq and scATAC-seq (Supplementary Figure S9)."

Figure R1. Clustering of B cell subpopulation based on the joint pattern of markers CD19, CD23, and CD69. The numbers represent the Hellinger distance.

Comment 2: There is also no information about the patient demographics, immunophenotype, genetic data on this CLL donor. CLL cells can show increased clonal heterogeneity, and the question is whether the subpopulations that the authors identified in CLL B cells have anything to do with the genetic heterogeneity (genetic subclonality) or even immunophenotypic heterogeneity.

Response: *Thank you for your comments. We have no access to the genotyping data, so we can not obtain any results related to genetic subclones. Based on the expression of surface markers, we see a distinct pattern among these B cell subpopulations that indicates immunophenotypic heterogeneity (see response to comment 1).*

Comment 3: I am not so sure about the purpose of Figure 4D. They compare the frequency of TP53 expressed cells among B1, B2, B7, and healthy donor's B cells. But what does this mean? There is no interpretation done. Also, the difference appears very small. 8.4% vs. 3-5%. It is not clear how this information is helping the authors argument.

Response: *Thank you for your comments. We have removed this figure since it is hard to give a clear interpretation.*

Comment 4: The paper does not discuss or compare the performance with another method such as MAESTRO. At least this needs to be added as a discussion.

Response: *Thank you for your comments. We have discussed and cited the method MAESTRO in the introduction section. The following description is added to the introduction section:*

"Recently, several other methods like SOMatic (9), scAI (10), and MAESTRO (11) are developed to integrate scRNA-seq with scATAC-seq."

Comment 5: The figures are shown in dissociated orders. It is recommended that figures are shown in order of how it is discussed in the text.

Response: *Thank you for your comments. We have reordered the figures to make them consistent with the order discussed in the text.*

Comment 6: Figure 4F and 4H has no unit for the x and y-axis.

Response: *Thank you for your comments. We have added units for the x and y-axis.*

Reviewer #2 (Bioinformatics of scTechnologies in cancer):

This manuscript presents sc-compReg, a method for identifying differential regulatory interactions between two cell populations based on scRNA-seq and scATAC-seq data. The problems of regulatory network inference and differential network analysis are well established with clear motivations in systems biology. This work is relevant to the increasing availability of single-cell multi-omics data sets combining scRNA-seq with scATAC-seq, for example, that present the opportunity to improve on existing algorithms. The authors also contribute an implementation for the joint embedding and clustering of cells based on the integration of gene expression and chromatin accessibility profiles, which is a step common to several downstream analyses of multi-omics data. There are interesting results presented for the identification of TOX2 as a tumor-specific regulator in chronic lymphocytic leukemia that are supported by further analysis and related literature.

Response: Thank you for your positive comments. We have implemented the new method in R and C++ based on the reviewers' and editor's comments, so it took longer than expected. We apologize for keeping you waiting.

MAJOR COMMENTS

Comment 1: While the authors introduce broad categories of relevant literature in Figure 1A with a few citations, there is not any citation or discussion of existing research that is most relevant to this manuscript, i.e., in the category of the comparative analysis of gene regulatory networks (Figure 1A, lower right). There are existing algorithms for gene regulatory network inference based on various forms of gene expression data, including the combination of scATAC-seq and scRNA-seq (e.g., Building gene regulatory networks from scATAC-seq and scRNA-seq using Linked Self Organizing Maps, <https://journals.plos.org/ploscompbiol/article?id=10.1371/journal.pcbi.1006555>). There is also substantial literature regarding the application of differential network analysis for the identification of (differential) interactions in gene regulatory networks (e.g., Comparative assessment of differential network analysis methods. <https://academic.oup.com/bib/article/18/5/837/2562785> and DNF: A differential network flow method to identify rewiring drivers for gene regulatory networks, <https://www.sciencedirect.com/science/article/pii/S0925231220308638>). The authors must provide a detailed discussion of the literature and provide a convincing argument that their research represents an advance over existing methods. There is certainly the potential for novel contributions in the proposed statistical method and the integration of scRNA-seq with scATAC-seq data, but it is not clear that these improve the performance of identifying differential regulatory interactions in comparison to methods that already exist in the literature.

Response: Thank you for your comments. In light of your comments, we have discussed and cited the relevant literature in the Introduction section. Please see the following two descriptions:

"To study gene regulation in a cell population, a widely used approach is based on paired single cell expression and accessibility analysis, where, scRNA-seq and scATAC-seq ⁷ experiments were performed on two different samples of cells from the same cell population (Fig. 1A, upper right). Duren et al⁸ have discussed

Recently, several other methods like SOMatic⁹, scAI¹⁰, and MAESTRO¹¹ are developed to integrate scRNA-seq with scATAC-seq."

"The main methodological question is how to identify differential regulatory relations based on the four single cell data sets (1 scRNA-seq + 1 scATAC-seq, from each of two populations).

.....

Several methods have been developed to detect the differential network based on bulk RNA-seq data ¹²⁻¹⁴. However, detecting the differential regulatory network by integration and comparison of scRNA-seq and scATAC-seq is a novel problem and it is challenging because whether a TF is involved in the differential regulation of a TG may depend in complex ways on the expression of the TF and the accessibility of REs that may mediate the activity of the TF on the TG (see Fig. 2)."

Comment 2: In this context, the validation of the proposed likelihood test for the identification of differential regulatory interactions is not sufficient. The results shown in Figure 2D and 2E demonstrate that the AUC for predictions made by the proposed method increases as additional accessibility information is integrated with gene expression to

determine a transcription factor's regulatory potential. These are the primary results that establish the performance of this method before its application to chronic lymphocytic leukemia. However, all of these results are based on simulated data, and there is no comparison between sc-compReg and any other differential network analysis algorithms.

Response: Thank you for your comments. We have compared our methods with other methods. Please see details in response to comment 5.

Comment 3: * There needs to be further discussion in the differential regulatory network data simulation section of the methods in order to justify why the simulated gene expression, regulatory element accessibility, and regulatory networks are representative of biological data.

Response: Thank you for your comments. We have added the following description in the Methods section to justify the simulation process.

"We simulate the target gene expression by the formulation provided by the PECA model²¹. The expression of j -th TG on the h -th cell is simulated as

$$TG_{jh} = v_j + \sum_k \gamma_{kj} \left(\sum_i B_{ik} \beta_{ij} RE_{ic} \right) TF_{kh} + \varepsilon; \varepsilon \sim N(0, (0.5 v_j)^2)$$

Where the v_j is the baseline expression of j -th TG which is generated from a Gamma distribution $v \sim \text{gamma}(2,2)$. To simulate TG expression by PECA formulation, we require TF expression, RE accessibility, TF-RE binding affinity, RE-TG association, and TF-TG regulatory structure. We use a normal distribution to generate the TF expression and RE accessibility for different cells, where the mean of TF expression and RE accessibility is generated from a gamma distribution. We randomly select some TF and RE, and add some value to them to simulate the differential TF and RE. TF-RE motif binding affinity and RE-TG connections are generated from a Bernoulli distribution. TF-TG connection should be very sparse so it is simulated as a mixture of a normal distribution with zero mean and a point mass at zero. To simulate differential regulatory structure, we randomly select some non-zero TF-TG connection and set their values to zeros."

Comment 4: * Additionally, the performance of sc-compReg has to be measured against some form of literature curated or experimentally validated gene regulatory networks, using experimental scRNA-seq and scATAC-seq data.

Response: Thank you for your comments. We agree that using literature curated or experimentally validated gene regulatory networks to assess the method provides the most solid validation. However, performing this type of validation requires scRNA-seq and scATAC-seq data on two relevant conditions together with the mentioned ground truth gene regulatory networks for both the conditions. Unfortunately, we haven't found any available dataset to perform this analysis.

Comment 5: * While published algorithms may not directly identify differential regulatory interactions in scRNA-seq with scATAC-seq data, regulatory networks output by existing GRN inference algorithms could be used as input to existing differential network analysis methods in order to compare their predictions with those made by sc-compReg and with ground truth networks. Benchmarking results could be used to identify the best performing regulatory network inference (e.g. Benchmarking algorithms for gene regulatory network inference from single-cell transcriptomic data, <https://www.nature.com/articles/s41592-019-0690-6>) and differential network analysis (e.g., Comparative assessment of differential network analysis

methods, <https://academic.oup.com/bib/article/18/5/837/2562785>) algorithms, available data sets and ground truth regulatory networks, and evaluation metrics for the validation of sc-compReg performance. In a comparison of sc-compReg with a combination of GRN inference and differential network analysis algorithms that do not integrate scATAC-seq with scRNA-seq data, sc-compReg results could be shown both with and without incorporating accessibility information in the calculation of transcription factor regulatory potential, as in the current Figure 2, in order to demonstrate the degree to which an improvement in performance results from the proposed likelihood ratio calculation or the inclusion of accessibility information from scATAC-seq data.

Response: Thank you for your instructive comments. In light of your suggestion, we have compared our method to different combinations of GRN inference methods and differential network analysis methods. Based on benchmarking results you suggested, we choose the best performance 2 GRN methods (GENIE3 and PIDC) and 2 differential analysis methods (DiffK and DiffRank) to assess our method. We compare these 4 combinations with our method on 4 different scenarios. Please see the following description for more detail.

Figure R2. ROC curves of differential regulatory detection on three simple scenarios of simulation data: differential TF expression, differential RE accessibility and differential regulatory network structure.

“An alternative way of performing this analysis is inferring the gene regulatory networks in each condition first and then detect differential regulation by comparing the two networks. We also compare our method with any combinations of two gene regulatory network inference methods and two differential regulatory detection methods. Based on benchmarking literature of gene regulatory network inference from single cell data¹⁵ and differential regulatory network detection¹³, we choose GENIE3¹⁶ and PIDC¹⁷ as methods for network inference and choose diffK¹⁸ and diffRank¹⁹ as methods for network comparison. As a result, our method outperforms all those methods (Fig. 2 D-E).”

Comment 6: The description of the rigorous validation methodology and the claim that sc-compReg is well-validated using data collected from bone marrow mononuclear cells is misleading. This strategy is only applied to the joint embedding and clustering approach, and is not used to validate the proposed likelihood ratio statistic for differential regulation detection, which is the primary novel contribution of this paper.

Response: Thank you for your comments. The validation methodology is for joint clustering and embedding not for the comparative regulatory analysis. We apologize for the misleading description. We have made two changes to address this issue. First, we have revised the fifth paragraph in the introduction section as

"Before we perform comparative regulatory analysis, we need an important initial analysis step to obtain the clustering label for each of the single cells and match the subpopulations across two conditions. To ensure the robust performance of the initial analysis, we developed a rigorous validation methodology, which may be of independent interest in the evaluation of single cell analysis methods. To validate a single cell analysis method, we apply it to scRNA-seq and/or scATAC-seq samples from a heterogeneous population with constituent subpopulations that are already analyzed by bulk sample RNA-seq and/or ATAC-seq. We use the bulk sample profiles to compute surrogate "ground truth" labels to the cells in our samples. These labels can then be used to validate the clustering, embedding, and subpopulation matching in the initial analysis step of sc-compReg. Applying this strategy on single cell data from a BMMC population, we showed that the initial analysis step of the sc-compReg pipeline is well-validated."

The original description was:

"To ensure the robust performance of sc-compReg, we developed a rigorous validation methodology, which may be of independent interest in the evaluation of single cell analysis methods. To validate a single cell analysis method, we apply it to scRNA-seq and/or scATAC-seq samples from a heterogeneous population with constituent subpopulations that are already analyzed by bulk sample RNA-seq and/or ATAC-seq. We use the bulk sample profiles to compute surrogate "ground truth" labels to the cells in our samples. These labels can then be used to validate the various steps in sc-compReg. Applying this strategy on single cell data from a BMMC population, we showed that the components steps of the sc-compReg pipeline are well-validated."

We have also revised the opening sentence of the "Comparative analysis reveals tumor-specific B cell subpopulations in chronic lymphocytic leukemia" section:

"Having validated the methodology, we next test it in a real application, namely the comparison of single cells from the BMMC sample of a CLL donor to that from a healthy donor. "

The original description was:

"We apply our method on the comparison of single cells from the BMMC sample of a CLL donor to that from a healthy donor. "

Comment 7: * It is necessary to be more precise in the definitions of the input, output, and variables referenced in the statistical test for differential regulatory relations section of Methods.

Response: Thank you for your comments. We have added a description of input and output in the beginning of the "Statistical test for differential regulatory relations" part in the Method section.

"To test if the regulation of i -th TF to j -th TG is differential in two conditions, we propose the following method. This method takes the cell-level expression of TF and TG and subpopulation-level accessibility profile of RE from two conditions as input, and calculates a likelihood ratio and p -value for each pair of TF and TG in each subpopulation."

Comment 8: There isn't an explicit definition for the values of the input variables TFit and TGjt.

Response: We have added these definitions. TF_{it} represents the expression of i -th TF in t -th cell. TG_{jt} represents the expression of j -th TG in t -th cell.

Comment 9: It isn't clear from the definition of the output variable LR that a likelihood ratio statistic is computed for each TFRP and TG pair.

Response: Thank you for your comments. We have added a note to emphasize that the LR is defined for i -th TF and j -th TG pair.

Comment 10: The parameters T1 and T2 in the LR calculation are not defined.

Response: Thank you for your comments. To make it easier for readers to understand, and also to address your comment 14, we have deleted the cell number T1 and T2 in the likelihood formulation.

Comment 11: There are subtle changes in the notation of related variables that should be explained, such as the meaning of the difference between X , X_{ht} , and X_t and the difference between β_{c0} , β_{c1} and β_0 , β_1 . For example, in the linear model, I presume that X is a matrix of expression values for all genes across all cells. Is that correct? Then what is X_t in (M0) and (M1)?

Response: Thank you for your comments. Here X is a matrix of TFRP, each row represents a TF-TG pair and each column represents a cell. Here the h is the index of TF-TG pair and t is index of cell. To make the notation clear, we use a full subscript in the formulation. We use $TFRP_{ijt}^{(1)}$ to represent TFRP of i -th TF on j -th TG in t -th cell from condition 1.

Comment 12: The notation might be simplified by replacing the subscript h , which denotes the h -th regulation between the i -th TF to j -th TG, with the subscript ij .

Response: Thank you for your comments. We have removed the subscript h .

Comment 13: Additionally, the notation shown in the related definition of TFRP and LR in Figure 2A and Figure 2C should be consistent with the methods, and the caption of Figure 2 should either reference the related methods or separately define the variables shown in the figure.

Response: Thank you for your comments. We replaced the variables X and Y in the method section with TFRP and TG to make the notation in Figure 2 and the method consistent. We have added some captions in Figure 2 and also refer to the Method section for detail.

Comment 14: Please provide rationale for the formulae for the likelihoods in (M0) and (M1). The rationales are clear after viewing the formulae carefully but a sentence or two of explanation will be helpful for the reader. It does not seem necessary to number all the cells from 1 to T1+T2. It is probably easier for the reader if the cells in the two conditions are numbered individually from 1 to T1 and from 1 to T2. That way both (M0) and (M1) will have two products, making the correspondence and the differences between them more apparent to the reader.

Response: Thank you for your comments. In light of your suggestion, we have provided a rationale for the formulae for the likelihoods in (M0) and (M1). We have also removed the number of cells T1 and T2 from the formula to make it easier for readers to understand.

Comment 15: The proposed validation methodology is convincing for the evaluation of joint embedding and clustering, although it isn't clear based on Figure 3 and Supplementary Note 3 that CoupledNMF significantly outperforms the other algorithms. While the accuracy for cell types noted in Supplementary Note 3 is higher for CoupledNMF, the results shown in Supplementary Figure S7 indicate there are other cell types in which Cell Ranger and Seurat have higher accuracy. A statistical comparison such as a two sample Kolmogorov-Smirnov test could be used to report a p-value demonstrating that the distribution of accuracy achieved by CoupledNMF is significantly higher than the distribution of accuracy achieved by Cell Ranger and Seurat. Similarly, the claim that cells of the same ground truth label are placed near each other in the joint embedding would be more convincing if it were supported by a quantitative measurement of the distance between cells with the same label in addition to referencing the plot in Figure 3D.

Response: *Thank you for your comments. In light of your comments, we have applied a quantitative measurement to evaluate the joint t-SNE. Specifically, we calculate silhouette index to show that cells from the same cell types are closer to each other in our joint t-SNE than t-SNE from scRNA-seq or scATAC-seq alone. The following description is added to the main text:*

"Based on the t-SNE coordinates and the surrogate ground truths, we calculate the silhouette index for each cell to evaluate how similar the cell is to its own cluster compared to other clusters. The joint tSNE provides significantly higher silhouette index than separate t-SNEs from Cell Ranger (sign rank test, $p=5.7973e-171$ for scATAC-seq and $p=5.7588e-08$ for scRNA-seq)."

Indeed, Seurat and Cell Ranger achieve better performance than the CoupledNMF for some cell types. But here our goal is not to cluster the cells but to perform comparative regulatory analysis which requires a pure cell population. So the challenging part is to obtain a pure cell population for minor cell types like Ery and CD8. The Cell Ranger lose the CD8 cluster in scRNA-seq and Seurat has less than 20% accuracy on the Ery cluster in scATAC-seq. If we choose these two methods, we cannot obtain a reasonable result on the comparative regulatory analysis of Ery or CD8.

MINOR COMMENTS

Comment 16: The Coupled clustering and Algorithm of CoupledNMF sections of the methods are closely related to the prior publication "Integrative analysis of single-cell genomics data by coupled nonnegative matrix factorizations," <https://www.pnas.org/content/pnas/115/30/7723.full.pdf>. The definition of TFRP in the Statistical test for differential regulatory relations section is closely related to the prior publication "Modeling gene regulation from paired expression and chromatin accessibility data," <https://www.pnas.org/content/pnas/114/25/E4914.full.pdf>. As written, the Methods and the Result present the impression that these methods are proposed in this manuscript. The authors should change the language to make it clear that these two methods have already been published. For example, they could describe newly proposed and existing methods under separate headers, and add citations for equations referenced from existing publications in order to highlight the novel contributions of this paper.

Response: *Thank you for your comments. We have changed the language to made it clear.*

Comment 17: The order of the sections in Methods isn't consistent with the order in which the results are presented. It will be more clear to change the order of the methods,

specifically, to describe the new statistical method before presenting the coupled clustering algorithm. This change will also highlight the new statistical method.

Response: *Thank you for your comments. In light of your comments, we have changed the order of the Methods to emphasize the new statistical method.*

Comment 18: It is difficult to distinguish between the solid and dashed lines in Figure 2D, especially when the lines overlap with each other or the axes of the plot. It would improve the readability of Figure 2D to use different colors for each line, and either increase the size of the plots or change the scale of the axes to further separate the two lines from each other and the axes.

Response: *Thank you for your comments. We have revised the color and added more methods to compare based on Comment 5.*

Comment 19: References to the upper left, upper right, lower left, and lower right cells of Figure 1A can be further clarified by adding alphanumeric labels for each category (e.g. Figure 1A, i ; Figure 1A, ii)

Response: *Thank you for your comments. We have further clarified Figure 1A by adding alphanumeric labels for each category.*

Comment 20: The caption of Figure 4B should contain a reference to the sub-section of Methods describing the linking of subpopulations across conditions. This change will clarify the meaning of PCC and mapping score. An alternative is to include those definitions in the caption.

Response: *Thank you for your comments. We have added a reference to the sub-section of Methods.*

Reviewer #3 (Remarks to the Author):

Duren et al. presents a new computational method sc-compReg. This method aims to use single cell chromatin accessibility and gene expression experiments, performed across two conditions, to infer how regulation by transcription factors differs between the two conditions. This "differential regulation" is only meaningful in the context of the same cell type across different conditions. Therefore, differential regulation estimation requires as an input a set of annotated cells, linked across experimental conditions. The authors provide software for this purpose.

The analysis their method enables would be of interest within the research community and their example applying it to a chronic lymphocytic leukemia adequately demonstrates its utility. There are some details of the methodology and presentation that could be improved (see below), but the main limitation of this method is the choice to implement it in a closed scripting language. This choice has a number of consequences. Firstly, it compels Duren et al to reinvent a series of well established routines for clustering and integration of annotation across experiments. An attempt is made to demonstrate that the methods created to perform these tasks are comparable to other established methods, which at best demonstrate that their methods are "good enough". They certainly do not advance the state of the art in any way.

Probably the worst consequence of providing an implementation requiring proprietary software to run is that it will limit its uptake by the community. By and large, a computational method is only as useful as the software that implements it. Were their method implemented

in an open language such as python or R, I could see it becoming the standard tool for differential regulation analysis of single cell RNA/ATAC data. As it stands, it is unlikely to be used outside those with an extremely strong need to do this analysis and it will likely be abandoned as soon as an open implementation becomes available.

That said, there is nothing fundamentally flawed about this paper or the method it describes that I can see. It will enable researcher to perform a specific type of analysis that will likely be of use to some. It is just a great shame that it's use and uptake is likely to be severely limited by the details of the software implementation.

Response: *Thank you for your positive comments. We have implemented the new method in R and C++ based on the reviewers' and editor's comments, so it took longer than expected. We apologize for keeping you waiting.*

Major criticisms:

Comment 1: One of the major limitations of the proposed method is its implementation in a proprietary software language. As I do not have a matlab license I could not test the sc-compReg and this is likely to greatly limit the appeal and utility of their method.

Response: *Thank you for your comments. We have taken this comment very seriously and spent a lot of time and effort to implement our method in R. Previously, we developed MATLAB-based software, because the matrix operation of MATLAB is much faster than R. To speed up the calculation, we use C++ to implement the core functions and develop a user-friendly R package. Please see our package hosted at this GitHub page: <https://github.com/SUwonglab/sc-compReg>*

Comment 2: I am not convinced that a new clustering and annotation method is needed. There are already a number of well established methods and software packages for clustering of single cell expression/ATAC data (e.g. scanpy/Seurat). I suspect the real reason this wheel has been reinvented here is because of point 1, that their choice of scripting language meant that they can not easily take advantage of the existing ecosystem of analysis tools in R/python.

Response: *Thank you for your comments. We agree that there are so many clustering methods. But here our main contribution is not developing a clustering method but developing a comparative regulatory analysis method. Clustering is just one step of the initial analysis, and here we need to jointly analyze scRNA-seq and scATAC-seq. In this initial analysis step, we compare three clustering methods and the results show that our previously developed CoupledNMF method has a good performance on joint clustering, so we choose this method as a default clustering method in our package. Of course, users can choose different clustering methods to perform comparative regulatory analysis by using our sc-compReg package.*

Comment 3: Similarity, there are many methods to identify similar populations of cells across two experiments.

Response: *Thank you for your comments. If we only consider scRNA-seq data, you are right. There are many methods that can identify linked populations across two conditions. But here we are considering both 1) integration of scRNA-seq and scATAC-seq and 2) comparison with other experimental conditions. As far as our knowledge, none of the existing methods identify similar populations of cells across two conditions by integrating scRNA-seq and scATAC-seq.*

Comment 4: The authors do provide a comparison of their clustering routines to other methods (Fig 3) which I would describe as only mildly convincing. This comparison relies on a torturous definition of ground truth which itself in effect applies label transfer from FACs sorted bulk data to single cell data. It would be far better to rely on biologically motivated manual annotation of existing data (such as PBMCs) as the ground truth. Leaving these concerns to one side, I would not say that the new "CoupledNMF" method performed better than existing approaches. All in all, the comparison is probably enough to demonstrate that the approach in the manuscript is "good enough" in the absence of alternatives, but not to justify its use over established approaches.

Response: Thank you for your comments. In this paper, we are introducing a new comparative regulatory method *sc-compReg* rather than the *CoupledNMF*. We published *CoupledNMF* in 2018, you may read this paper from the following link <https://www.pnas.org/content/115/30/7723.short> . We agree that the the presented results are not enough to conclude that our method for joint clustering and embedding is better than others. However, it is not our aim to convince people to stop using Seurat and other methods and use our method. Instead, our purpose is to choose a well-justified default joint clustering method to generate the input data for the comparative regulatory analysis. We also agree that manual annotation is also a solid way to validate the clustering method. For scRNA-seq data, this is relatively straightforward since there are many known marker genes for these cell types. But for scATAC-seq data, the manual annotation is very difficult since we do not have so many known markers in chromatin.

Other comments:

Comment 5: The figures are very heavy on text/formula that cannot be understood with just the figure plus legend alone (e.g. Fig 2C). Such details would be better included in the text/methods where they should be explained in full.

Response: Thank you for your comments. Fig 2C is our main contribution and so it is better to keep it there. We have a full description in the "Statistical test for differential regulatory relations" in the Methods section.

Comment 6: It is not clear to me that H_0 is nested within H_1 , as is required for the likelihood ratio statistic to follow a chi-squared distribution. Could the fitting procedure on X be reformulated in terms of the difference between the two conditions? The null would then become $\beta=0$, with the alternative $\beta \neq 0$.

Response: Thank you for your comments. The H_0 and H_1 could be reformulated as:

$$H_0: (\beta_{10}, \beta_{11}, \sigma_1) - (\beta_{20}, \beta_{21}, \sigma_2) = 0$$

$$H_1: (\beta_{10}, \beta_{11}, \sigma_1) - (\beta_{20}, \beta_{21}, \sigma_2) \neq 0$$

Comment 7: The change in TP53 expression level between populations is not particularly striking and while I'm sure it is statistically significant this is insufficient to decide which populations are tumor. Does the sample being considered possess somatic mutations in TP53?

Response: Thank you for your comments. We have removed this figure since, as you say, it is not very striking and hard to interpret clearly.

Reviewers' Comments:

Reviewer #1:

Remarks to the Author:

The authors analyzed the gene expression differences of CD5, CD19, CD23, CD69, and CD71 among the three B cell subpopulations detected in CLL samples (B1, B2, and B7). They found that the expression of CD5 and CD71 was too low to perform correlation which was likely due to the dropout from scRNA seq. However, B7 subpopulations had the most distinct expression compared to normal B cells. Based on this finding, authors claimed that B7 is likely representative of CLL cells.

It was somewhat disappointing that CD5 expression was not evaluable from scRNA seq data due to the dropout. Since CD5+CD23+CD19+ is the hallmark of CLL cells.

In response to my comment #2, the authors stated that they had no genotype data so was unable to correlate. However, authors can perform copy number analysis based on scRNA-seq data, which might identify copy number abnormalities that maybe unique to CLL cells. For example, 50% of CLL carries del13q and other CLL may carry, Tris 12, del11q, or del17p and others. If B7 sub-population carries one of these copy number alterations, it will further strengthen the argument that B7 population represents CLL cells. Since the last part of this paper really relies on the fact that B7 population is CLL, in my opinion, it is really important to determine the identify of B7 population. Therefore, I recommend performing copy number analysis with an attempt to further strengthen the argument that B7 population is CLL cells.

Reviewer #2:

Remarks to the Author:

Thank you to the authors for systematically addressing comments on the first revision. In particular, the ROC curve results added to Fig. 2 with comparisons between sc-compReg of other published algorithms for GRN inference and differential network analysis algorithms more clearly establish that sc-compReg can outperform existing methods by integrating scATAC-seq with scRNA-seq data. The updated notation in the formulation of the method also helps clarify details of the input and definition of variables.

I have two major comments.

My comment #4 on the first version of the manuscript was that the performance of sc-compReg has to be measured against some form of literature curated or experimentally validated gene regulatory networks. In response, the authors have said that they have not found scRNA-seq and scATACseq datasets on two relevant conditions together with the mentioned ground truth gene regulatory networks for both the conditions.

I recognize these challenges. Nevertheless, establishing that sc-compReg is superior to competing techniques on experimental data is critical for the publication of the manuscript. Otherwise, there is too heavy a reliance on synthetic data. However, it is still possible to support the biological relevance and improvement of sc-compReg predictions. For example, demonstrating that predictions made by sc-compReg are more informative than existing computational methods might be done using the same CLL data set. In order to further establish the biological relevance of sc-compReg predictions, I suggest that the following strategy:

1. Process the CLL scRNA-seq data using the same combinations of GRN inference and differential network analysis algorithms as in the simulated data.
2. As a gold standard network, use a reference database of ChIP-seq binding motif enrichment such as cisTarget (which is used by SCENIC to process the output of GENIE3 for GRN inference) or Gene Transcription Regulation Database (GTRD).

3. Compare all predicted TF-TG regulatory relationships with TF-TG pairs in the gold standard network. Use precision-recall curves and areas under these curves across to measure and compare performance of different methods.

4. Add a figure similar to Fig. 2 that shows differential TF-TG relationships identified by sc-compReg are more accurate than existing methods with respect to experimental ChIP-seq data.

My comment #3 on the first version of the manuscript was that there needs to be further discussion to justify why the simulated gene expression, regulatory element accessibility, and regulatory networks are representative of biological data. In response, the authors have added more details and explanatory text.

I thank them for adding more details on the generation of simulated data. However, a stronger argument should still be made in the text for why this simulated data is representative of biological data. As written, the new text is almost entirely an explanation in words of the equations rather than a rationale for why these equations are appropriate for single-cell data. For example, why is a normal distribution appropriate for TF expression and RE accessibility? Should the simulated data include an approximation for noise caused by drop-out inherent to experimental single-cell data sets? Why is this the correct formulation of simulated data when performance using this data set is the primary measurement against which sc-compReg is compared to alternative existing methods?

Reviewer #3:

Remarks to the Author:

The authors have taken the reviewers comments seriously and responded to them well. They should be particularly commended for having taken the effort to reimplement their software in a more widely used language. I am happy to recommend the revised manuscript for publication.

Reviewer #1 (Expertise: The use of sc technologies in blood cancers):

The authors analyzed the gene expression differences of CD5, CD19, CD23, CD69, and CD71 among the three B cell subpopulations detected in CLL samples (B1, B2, and B7). They found that the expression of CD5 and CD71 was too low to perform correlation which was likely due to the dropout from scRNA seq. However, B7 subpopulations had the most distinct expression compared to normal B cells. Based on this finding, authors claimed that B7 is likely representative of CLL cells. It was somewhat disappointing that CD5 expression was not evaluable from scRNA seq data due to the dropout. Since CD5+CD23+CD19+ is the hallmark of CLL cells.

Response: Thank you for your comments.

Comment 1: In response to my comment #2, the authors stated that they had no genotype data so was unable to correlate. However, authors can perform copy number analysis based on scRNA-seq data, which might identify copy number abnormalities that maybe unique to CLL cells. For example, 50% of CLL carries del13q and other CLL may carry, Tri 12, del11q, or del17p and others. If B7 sub-population carries one of these copy number alterations, it will further strengthen the argument that B7 population represents CLL cells. Since the last part of this paper really relies on the fact that B7 population is CLL, in my opinion, it is really important to determine the identify of B7 population. Therefore, I recommend performing copy number analysis with an attempt to further strengthen the argument that B7 population is CLL cells.

Response: Thank you for your comments. In the light of your suggestion, we did copy number analysis based on a recently published method copyKAT. Supplementary Figure S10A shows the B2 and B7 are 2.04 and 1.86 fold enriched (odd ratio) in the predicted aneuploid cells, while the odd ratio of healthy B cells and B1 are 0.22 and 1.22.

Figure R1. Odd ratios of copyKAT predicted aneuploid cells' enrichment in B cell subpopulations. The error bar represents the 95% confidence interval.

We compare the distribution of predicted copy number between CLL and healthy B cells. We obtained 53, 75, and 66 loci which have significant copy number difference in B1, B2, and B7 compared to healthy B cells. All the significant loci in B1 are either overlapped with B2 or B7. B2 has 19 unique significant loci and B7 has 15 significant loci (Supplementary Figure S10B). The results from copy number analysis support that the B1 is more similar to normal B cell and B2 and B7 are different from normal B cells.

Figure R2. The Venn diagram of overlapping of significant differential copy number locus between three B cell subpopulation in CLL.

Reviewer #2 (Bioinformatics of scTechnologies in cancer):

Thank you to the authors for systematically addressing comments on the first revision. In particular, the ROC curve results added to Fig. 2 with comparisons between sc-compReg of other published algorithms for GRN inference and differential network analysis algorithms more clearly establish that sc-compReg can outperform existing methods by integrating scATAC-seq with scRNA-seq data. The updated notation in the formulation of the method also helps clarify details of the input and definition of variables. I have two major comments.

Response: Thank you for your comments.

Comment 1: My comment #4 on the first version of the manuscript was that the performance of sc-compReg has to be measured against some form of literature curated or experimentally validated gene regulatory networks. In response, the authors have said that they have not found scRNA-seq and scATACseq datasets on two relevant conditions together with the mentioned ground truth gene regulatory networks for both the conditions. I recognize these challenges. Nevertheless, establishing that sc-compReg is superior to competing techniques on experimental data is critical for the publication of the manuscript. Otherwise, there is too heavy a reliance on synthetic data. However, it is still possible to support the biological relevance and improvement of sc-compReg predictions. For example, demonstrating that predictions made by sc-compReg are more informative than existing computational methods might be done using the same CLL data set. In order to further establish the biological relevance

of sc-compReg predictions, I suggest that the following strategy: 1. Process the CLL scRNA-seq data using the same combinations of GRN inference and differential network analysis algorithms as in the simulated data. 2. As a gold standard network, use a reference database of ChIP-seq binding motif enrichment such as cisTarget (which is used by SCENIC to process the output of GENIE3 for GRN inference) or Gene Transcription Regulation Database (GTRD). 3. Compare all predicted TF-TG regulatory relationships with TF-TG pairs in the gold standard network. Use precision-recall curves and areas under these curves across to measure and compare performance of different methods. 4. Add a figure similar to Fig. 2 that shows differential TF-TG relationships identified by sc-compReg are more accurate than existing methods with respect to experimental ChIP-seq data.

Response: Thank you for your comments. As you suggested, we process the CLL and healthy scRNA-seq data by the same combinations of GRN inference methods and differential network analysis methods as in the simulation data. We use the knowledge-based regulatory network database Regnetwork (<http://regnetworkweb.org/>) as ground truth to validate the networks. We perform differential regulatory analysis between the three B cell subpopulations and the healthy B cells. The Supplementary Figure S11 shows the precision-recall curves and areas under curves. As the collected ground truth is incomplete and also it contains regulation from non-relevant cellular context, the AUC of all methods are low. But it is still sufficient to compare the performance of different methods. Our method has the best performance in all the three B cell subpopulations.

Figure R3. Validation of differential regulatory network between three B cell subpopulation in CLL and healthy B cell. The ground truth is from the knowledge-based transcriptional regulatory network database Regnetwork. The figure shows the precision-recall curves. The lower right figure is after merging the prediction of 3 comparisons.

Comment 2: My comment #3 on the first version of the manuscript was that there needs to be further discussion to justify why the simulated gene expression, regulatory element accessibility, and regulatory networks are representative of biological data. In response, the authors have added more details and explanatory text. I thank them for adding more details on the generation of simulated data. However, a stronger argument should still be made in the text for why this simulated data is representative of biological data. As written, the new text is almost entirely an explanation in words of the equations rather than a rationale for why these equations are appropriate for single-cell data. For example, why is a normal distribution appropriate for TF expression and RE accessibility? Should the simulated data include an approximation for noise caused by drop-out inherent to experimental single-cell data sets? Why is this the correct formulation of simulated data when performance using this data set is the primary measurement against which sc-compReg is compared to alternative existing methods?

Response: Thank you for comments. We have revised our description to justify the simulation. Please see our response to your three specific questions.

1) **Why is a normal distribution appropriate for TF expression and RE accessibility?** Here we are simulating a “real” gene expression level. The observed data includes drop-out. We are assuming that the “real” expression level of a gene in a pure cell population should follow a normal distribution. We tested this assumption in a deep sequenced scRNA-seq data. In retinoic acid induced mESC differentiation data (Duren et al 2018), we use Kolmogorov-Smirnov test to examine if the \log_2 transformed expression data (zeros are removed) of 21973 genes are from normal distributions. As a results, 21459 (97.66 %), 21432 (97.54%), and 19980 (90.93%)) genes are not significantly different from a normal distribution ($P > 0.05$) in cluster 1, 2 and 3 respectively. The result on this data supports our assumption of normal distribution. This should be similar in chromatin accessibility. We also assume that the average expression level of a gene follows a Gamma distribution. We use the ENCODE data to justify this. The ENCODE data contain expression of 18547 protein coding genes on 241 cellular context. The Supplementary Figure S16 shows that Gamma is a good fit for the distribution of average gene expression.

Figure R4. Distribution of average gene expression follows a Gamma distribution.

2) **Should the simulated data include an approximation for noise caused by drop-out inherent to experimental single-cell data sets?** For each expression, we generate a random variable from standard uniform distribution $U(0,1)$. The expression level will be dropped to zero if this generated number is less than $\frac{1}{1+e^{-\kappa \log(E)-\lambda}}$. A gene with lower expression level is more likely to be dropped out.

3) **Why is this the correct formulation of simulated data when performance using this data set is the primary measurement against which sc-compReg is compared to alternative existing methods?** It is a fair comparison because the model used to generate the simulation data and the model to be evaluated are totally different. The formulation of simulated data is based on PECA model, and the expression of target genes are determined by the combined effect of multiple TFs. But model in sc-compReg just focused on the pairwise relations not considering the joint effect of multiple TFs.

Reviewer #3

The authors have taken the reviewers comments seriously and responded to them well. They should be particularly commended for having taken the effort to reimplement their software in a more widely used language. I am happy to recommend the revised manuscript for publication.

Response: *Thank you for recommending our manuscript for publication.*

Reviewers' Comments:

Reviewer #1:

Remarks to the Author:

Thank you for performing CopyKAT analysis. However, the authors only showed the correlation between B-cell subpopulation and total copy number abnormality burden. They did not show specific copy number abnormalities that we should expect for CLL cells such as but not limited to del13q, del11q, trisomy12, del17p. Did B7 population had any of those copy number changes?

Also, I think the authors did not submit supplemental figures in this round of submission. I was not able to see the actual Supp Figure S10B that authors mentioned.

Reviewer #2:

Remarks to the Author:

The authors have responded satisfactorily to my second round of comments. It is disappointing but not surprising that the precision-recall results for all algorithms are poor. It is somewhat reassuring that the methods proposed by the authors are the best. I support the publication of the manuscript in this form.

Reviewer #1 (Expertise: The use of sc technologies in blood cancers):

Thank you for performing CopyKAT analysis. However, the authors only showed the correlation between B-cell subpopulation and total copy number abnormality burden. They did not show specific copy number abnormalities that we should expect for CLL cells such as but not limited to del13q, del11q, trisomy12, del17p. Did B7 population had any of those copy number changes?

Also, I think the authors did not submit supplemental figures in this round of submission. I was not able to see the actual Supp Figure S10B that authors mentioned.

Response: *Thank you for your comments. We identified 98 loci which have significant copy number differences to healthy B cells. In light of your suggestion, we have listed them in the supplementary Table S1.*

We apologize for the incomplete submission. Please see all the supplementary figures and table in the this new submission.